# SPEAR: A Unified SSL Framework for Learning Speech and Audio Representations

Xiaoyu Yang[1]   Yifan Yang[4]   Zengrui Jin[3]   Ziyun Cui[2,3]   Wen Wu[2]   Baoxiang Li[2]   Chao Zhang[2,3]
Phil Woodland[1]

## Abstract

Self-supervised learning (SSL) has significantly advanced acoustic representation learning. However, most existing models are optimised for either speech or audio event understanding, resulting in a persistent gap between these two domains. We address this gap with SPEAR (SPEech and Audio Representations), a self-supervised framework that distils complementary knowledge from a speech-focused SSL teacher and a general-audio SSL teacher into a single unified model. SPEAR applies multi-codebook vector quantisation to continuous teacher representations to produce fine-grained discrete tokens that capture both semantic and acoustic information. To effectively integrate these heterogeneous representations, SPEAR jointly predicts them given a masked input with an asymmetric pre-training loss. We further improve robustness in complex sound scenes through a novel token mixing mechanism. Extensive experiments demonstrate that SPEAR consistently outperforms existing unified speech and audio models. SPEAR establishes a new state-of-the-art on the SUPERB benchmark, surpassing WavLM Large on 12 of 15 tasks, while achieving competitive performance on the HEAR benchmark. These results position SPEAR as a versatile foundation for general-purpose speech and audio representation learning. The code and pre-trained models will be released[1].

---

[1]Department of Engineering, University of Cambridge, Cambridge, UK. [2]Shanghai Artificial Intelligence Laboratory, Shanghai, China. [3]Tsinghua University, Beijing, China. [4]Shanghai Jiao Tong University, Shanghai, China.. Correspondence to: Xiaoyu Yang <xy316@cam.ac.uk>.

*Proceedings of the 43rd International Conference on Machine Learning*, Seoul, South Korea. PMLR 306, 2026. Copyright 2026 by the author(s).

[1]https://huggingface.co/collections/marcoyang/spear-encoders

## 1. Introduction

The pursuit of general-purpose foundation models has driven transformative progress in natural language processing (Brown et al., 2020) and computer vision (Kirillov et al., 2023). In contrast, acoustic signal processing remains fragmented into three traditionally disjoint domains: speech, audio events, and music. Speech processing focuses on linguistic and paralinguistic information, typically treating non-speech sounds as noise to be suppressed. Conversely, audio event processing aims to classify acoustic events, and it often collapses speech into coarse categories, losing semantic content (*e.g.*, "a man speaking"). Music processing lies at the intersection, requiring both acoustic event recognition for instrumentation and linguistic understanding for lyrics and prosody.

Crucially, real-world environments present **complex sound scenes** containing multiple, often overlapping, sound sources. Treating distinct domains in isolation neglects their intrinsic synergy: environmental sounds provide essential context for speech perception, while speech content can disambiguate sound events (Bregman, 1990). Consequently, reliance on specialised models limits holistic speech and audio understanding, motivating the need for a unified representation space. For clarity, we use the term "**speech**" to denote speech-centric tasks that rely on linguistic and paralinguistic information, and "**audio**" to collectively refer to tasks emphasising environmental sounds and music throughout the remainder of this paper.

Against this backdrop, self-supervised learning (SSL) has emerged as a powerful paradigm for learning generic representations. While its effectiveness has been demonstrated independently in speech (Baevski et al., 2020; Hsu et al., 2021; Baevski et al., 2022; Chiu et al., 2022) and audio (Chen et al., 2023; Huang et al., 2022; Pepino et al., 2025; Dinkel et al., 2024; Chen et al., 2024a) domains (Yang et al., 2021; Turian et al., 2022), constructing a single foundation model that generalises across both remains difficult due to divergent inductive biases. Speech SSL models typically utilise coarse-grained units to capture phonetic structure, which are insufficient for modelling the irregular, complex spectro-temporal patterns of general audio. Conversely, audio SSL

models tend to focus on low-level acoustic details instead of high-level semantics required for speech-centric tasks. To bridge this gap, recent studies (Yang et al., 2025; Chang et al., 2025) have explored distilling complementary knowledge from domain-specific teachers into a single model.

To address this lack of unification, we propose **SPEAR** (SPEech and Audio Representations), a unified self-supervised learning framework for jointly learning speech and general audio representations. SPEAR is motivated by the hypothesis that masked prediction over fine-grained discrete tokens can serve as a shared learning interface for self-supervision across both domains. Concretely, SPEAR employs multi-codebook vector quantisation (MVQ) (Guo et al., 2023) to derive fine-grained discrete tokens from intermediate representations of existing speech- and audio-oriented SSL models, which are then used as supervision targets in a masked token prediction objective. Compared to conventional coarse-grained discretisation schemes (*e.g.*, $k$-means), MVQ preserves substantially richer acoustic and temporal detail, making the resulting tokens suitable for both speech-centric and general audio modelling. By integrating masked token prediction with knowledge distillation, SPEAR enables the student encoder to inherit complementary strengths from domain-specific teachers, while simultaneously encouraging the discovery of both high-level semantic structure and fine-grained acoustic detail directly from data. SPEAR is pre-trained on both speech and audio data, where the model jointly predicts two sets of MVQ tokens distilled from the respective domain-specific teachers. To ensure balanced learning and effective fusion across domains, we further introduce an **asymmetric dual-domain objective** that explicitly accounts for their differing characteristics. Finally, to improve robustness in complex sound scenes with overlapping sound sources, we propose a **token mixing mechanism** that constructs augmented training samples by stochastically combining supervision targets from multiple sources.

Our contributions can be highlighted as follows:

- We propose SPEAR, a state-of-the-art SSL framework that bridges the gap between speech and general audio representation learning by distilling complementary knowledge from domain-specific teachers into a single unified model.

- We introduce a novel domain fusion strategy to integrate the heterogeneous representations from two teachers, where we jointly predict the fine-grained MVQ tokens derived from the continuous representation space of the teachers, given a masked input in an asymmetrical manner.

- We propose a token mixing mechanism that stochastically fuses the targets from multiple sources, yielding

consistent improvements on complex sound scenes.

## 2. Related Work

**Speech SSL and Audio SSL**  The field of speech and audio processing covers a wide range of tasks (Gold et al., 2011), such as automatic speech recognition (ASR), speaker verification (SV), music instrument classification, and audio tagging (AT). Some tasks require high-level semantic understanding (*e.g.*, ASR), while others focus on low-level acoustic detail (*e.g.*, AT, pitch estimation). This poses challenges in learning a unified representation space for both the speech and the general audio domains (Bengio et al., 2013). SSL has been widely adopted for learning generic representations in both the speech (Baevski et al., 2020; Hsu et al., 2021; Chen et al., 2022) and the audio domains (Huang et al., 2022; Dinkel et al., 2024). Yet, existing SSL methods are typically domain-specific and transfer poorly to the other domain, and a unified framework for both domains remains underexplored. While bootstrapping approaches (Grill et al., 2020) have shown promise in speech (Baevski et al., 2022; 2023) and audio (Chen et al., 2024a; Li et al., 2024) independently, directly applying such objectives (*e.g.*, data2vec) to joint pre-training on speech and audio data degrades performance by a large margin (Chang et al., 2025). Similarly, Dasheng (Dinkel et al., 2024) applied an MAE (He et al., 2022) objective to large-scale speech and audio data, but only showed limited capability on speech tasks requiring semantic understanding.

**Masked-Token Prediction and Knowledge Distillation**  Work most closely related to SPEAR includes Encodec-MAE (Pepino et al., 2025), MT2KD (Yang et al., 2025) and USAD (Chang et al., 2025). EncodecMAE performs masked-token prediction on Encodec RVQ tokens (Défossez et al., 2023) for audio representation learning. However, it overlooks the hierarchy of RVQ tokens and is only applicable to the general audio domain, showing weak generalisation on speech tasks. In contrast, SPEAR uses non-hierarchical MVQ tokens as prediction targets and generalises to the speech domain. MT2KD (Yang et al., 2025) was the first to build a unified encoder by performing knowledge distillation (KD) on three teachers specialising in ASR, SV, and AT, using L1 and cosine similarity. However, the resulting model is constrained to the three speech and audio processing tasks since the teacher models use supervised training. USAD jointly distils two SSL teachers specialising in speech and the general audio domain into a single unified encoder. Similar to MT2KD, USAD employs feature matching losses to align the student representations with those of the teacher layer-wise. While both methods learn a unified representation space, their performance is constrained by the teacher models since they are purely mimicking the teacher representations. SPEAR fundamen-

tally differs from this paradigm by reformulating domain fusion as a masked token prediction task over a discrete interface. Specifically, SPEAR combines KD with a well-established masked token prediction pretext task instead of relying solely on feature matching losses. This encourages the model to learn contextualised representation by discovering semantic and acoustic structures in the data rather than purely mimicking the teacher. Additionally, SPEAR considers the domain mismatch between speech and general audio, and introduces an asymmetrical pre-training loss to cope with it. Furthermore, while prior distillation frameworks (*e.g.*, MT2KD and USAD) are not explicitly optimised for complex, overlapping sound scenes, SPEAR directly targets multi-source representation fidelity in such environments by introducing a novel token mixing mechanism.

**Speech and Audio Encoders for Audio LLMs**   Other work related to SPEAR includes SALMONN (Tang et al., 2024), Audio Flamingo 3 (Ghosh et al., 2025) and Qwen-omni (Xu et al., 2025). SALMONN found that it is important to combine a speech specialist with an audio-event specialist for a comprehensive audio perception capability in multi-modal large language models (LLMs). Audio Flamingo 3 and Qwen-omni train an audio encoder with text LLMs end-to-end, requiring a vast amount of paired data and computation resources. However, SPEAR relies purely on unlabelled data for learning unified speech and audio representations, since the training labels are derived from domain-specific SSL teachers trained without supervision.

## 3. SPEAR

### 3.1. Multi-codebook Vector Quantisation

We hypothesise that masked-token prediction can serve as a unified SSL objective for both speech and general audio, provided the discrete tokens are sufficiently fine-grained to retain critical semantic and acoustic detail from both domains. In SPEAR, we employ multi-codebook vector quantisation (MVQ) (Guo et al., 2023), a trainable vector quantisation method to generate fine-grained discrete targets for our masked prediction SSL objective. MVQ was originally proposed to compress high-dimensional feature vectors for storage optimisation. To the best of our knowledge, the application of MVQ in the context of SSL pre-training has not been explored. MVQ utilises $N$ parallel codebooks, each containing $K$ trainable code vectors. Given an input feature vector $\boldsymbol{x} \in \mathbb{R}^d$, MVQ encodes it into a tuple of $N$ discrete tokens, i.e. $\boldsymbol{z} = \mathrm{Encode}(\boldsymbol{x}; \mathcal{Q}) = (z_1, \ldots, z_N)$. Each token $z_n$ is an integer index in $[0, K-1]$, specifying which code to select from the $n$-th codebook. These selected vectors can then be used to approximate the original feature vector $\boldsymbol{x}$ via a direct-sum scheme (Barnes & Watkins, 1995).

Intuitively, MVQ partitions a feature space into $N$ distinct

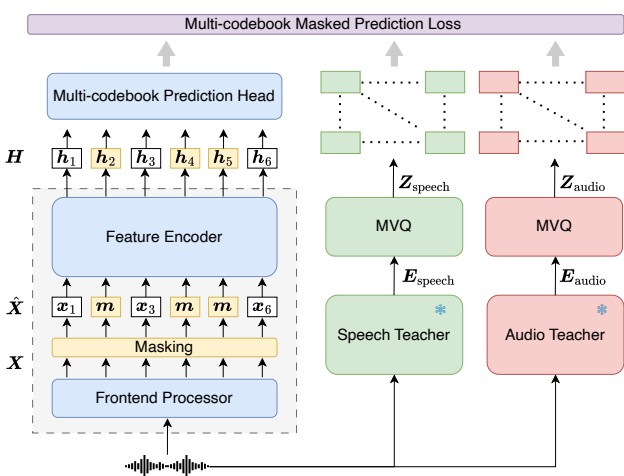

*Figure 1.* The SPEAR framework for dual-domain pre-training. Teacher models are frozen. For single-domain pre-training, only one teacher from the corresponding field is employed for generating pre-training targets. After pre-training, the components in the grey box are retained as the encoder.

subspaces, each governed by a single codebook. The multi-codebook design produces far more fine-grained representations than coarse methods, *e.g.*, $k$-means, as the number of representable states grows exponentially to $K^N$. Compared to other multi-codebook quantisation methods such as RVQ (Défossez et al., 2023), the codebooks in MVQ are non-hierarchical and independent. MVQ is trained by minimising the reconstruction error, supplemented by a diversity loss to encourage uniform code usage within each codebook. For a complete description of MVQ encoding and training mechanisms, we refer readers to the original paper (Guo et al., 2023) and the extended summary in Appendix A.

### 3.2. Unified Speech and Audio Representation Learning

We first elaborate on how to leverage MVQ tokens for representation learning in a single teacher scenario, and then extend it to unified representation learning over both speech and audio domains.

#### 3.2.1. MVQ-BASED MASKED TOKEN PREDICTION

In the single teacher scenario, the pre-training objective is to train a student encoder $\mathcal{S}$ to predict fine-grained MVQ tokens extracted from a pre-trained SSL teacher model $\mathcal{T}$ in a masked-token prediction manner. Since the teacher used for generating the pre-training targets was trained without any labelled data, SPEAR is treated as an SSL approach. An illustration of the overall framework is shown in Figure 1.

The student encoder $\mathcal{S}$ consists of a frontend processor and a feature encoder $\mathcal{F}$. The frontend processor converts the raw input waveform $\boldsymbol{w}$ into frame-level representations $\boldsymbol{X} = \{\boldsymbol{x}_1, \ldots, \boldsymbol{x}_T\}$ of length $T$. A masking operation is

applied to $\boldsymbol{X}$ by randomly sampling a set of frames $\mathcal{M}$ and replacing $\{\boldsymbol{x}_t | t \in \mathcal{M}\}$ with a learnable mask embedding $\boldsymbol{m}$, creating the masked input $\hat{\boldsymbol{X}}$. The feature encoder $\mathcal{F}$ then processes $\hat{\boldsymbol{X}}$ to produce a sequence of contextualised representations $\boldsymbol{H} = \{\boldsymbol{h}_1, \ldots, \boldsymbol{h}_T\}$, where $\boldsymbol{h}_t \in \mathbb{R}^d$.

To generate the prediction targets, $\boldsymbol{w}$ is also fed into the teacher model $\mathcal{T}$, producing a sequence of frame-level representations $\boldsymbol{E} = \mathcal{T}(\boldsymbol{w}) = \{\boldsymbol{e}_1, \ldots, \boldsymbol{e}_T\}$. We assume the teacher and student models share the same frame rate[2]. These representations are then quantised frame-by-frame using a pre-trained MVQ quantiser $\mathcal{Q}$ to produce a sequence of fine-grained discrete tokens $\boldsymbol{Z} = \{\boldsymbol{z}_1, \ldots, \boldsymbol{z}_T\}$ as the pre-training targets, where $\boldsymbol{z}_t = \text{Encode}(\boldsymbol{e}_t; \mathcal{Q})$.

The student model is trained to predict the target tokens $\boldsymbol{Z}$ from the contextualised representations $\boldsymbol{H}$. The multi-codebook masked prediction loss is formulated as the sum of $N$ independent prediction losses, one for each codebook in the MVQ quantiser. Each of these losses is a cross-entropy objective calculated over all frames, with an adjustable weight $\alpha$ for masked and unmasked frames[3]:

$$\mathcal{L}_{\text{single}}(\boldsymbol{H}, \boldsymbol{Z}) = \frac{1}{N} \sum_{n=1}^{N} [\alpha \mathcal{L}_{n,m} + (1 - \alpha)\mathcal{L}_{n,u}] \quad (1)$$

$$\mathcal{L}_{n,m} = \sum_{t \in \mathcal{M}} -\log p_n(z_{t,n} \mid \boldsymbol{h}_t) \quad (2)$$

$$\mathcal{L}_{n,u} = \sum_{t \notin \mathcal{M}} -\log p_n(z_{t,n} \mid \boldsymbol{h}_t) \quad (3)$$

where $\mathcal{L}_{n,m}$ and $\mathcal{L}_{n,u}$ are the loss on masked and unmasked frames for the $n$-th codebook, respectively. $p_n(z_{t,n}|\boldsymbol{h}_t)$ is the predicted probability of the correct token $z_{t,n}$ at time $t$ for the $n$-th codebook, generated by $N$ independent linear prediction heads on top of the feature encoder.

### 3.2.2. ASYMMETRICAL DUAL-DOMAIN PRE-TRAINING

The framework is extended to dual-domain pre-training on a mixture of speech and general audio data for learning unified representations of both domains. Specifically, we employ two expert teacher models, $\mathcal{T}_{\text{speech}}$ and $\mathcal{T}_{\text{audio}}$, along with their corresponding pre-trained MVQ quantisers, $\mathcal{Q}_{\text{speech}}$ and $\mathcal{Q}_{\text{audio}}$. For each input waveform, the teacher representations $\boldsymbol{E}_{\text{speech}}$ and $\boldsymbol{E}_{\text{audio}}$ are extracted. Two sets of MVQ tokens $\boldsymbol{Z}_{\text{speech}}$ and $\boldsymbol{Z}_{\text{audio}}$ are obtained by applying the corresponding quantiser on the corresponding representations.

Since the teacher models are domain-specific, simply computing losses against both targets regardless of the domain of the input data could be suboptimal. Therefore, we design three dual-domain pre-training strategies:

---

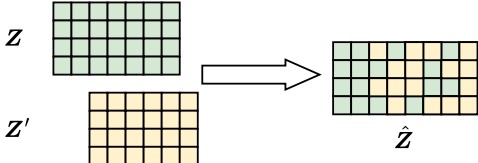

*Figure 2.* A token mixing example with $\beta = 0.6$ and $\tau = 2$.

- JOINT: Each sample $\boldsymbol{w}$ induces two losses, computed against $\boldsymbol{Z}_{\text{speech}}$ and $\boldsymbol{Z}_{\text{audio}}$, regardless of its domain.

- DISJOINT: Each training input $\boldsymbol{w}$ only induces one loss, computed against the targets generated by the teacher from the same domain as $\boldsymbol{w}$.

- ASYMMETRICAL: For speech data, loss is only computed against $\boldsymbol{Z}_{\text{speech}}$. For audio data, loss is computed against both $\boldsymbol{Z}_{\text{speech}}$ and $\boldsymbol{Z}_{\text{audio}}$.

In SPEAR, the ASYMMETRICAL strategy was adopted because it yields the best performance (see Section 5.4 for a comparison of the three strategies). That said, the speech tokens $\boldsymbol{Z}_{\text{speech}}$ are used as universal prediction targets for all input data, whereas the audio tokens $\boldsymbol{Z}_{\text{audio}}$ are only used as targets for general audio input. The asymmetrical dual-domain pre-train objective is formulated as follows:

$$\mathcal{L}_{\text{dual}} = \mathcal{L}_{\text{single}}(\boldsymbol{H}, \boldsymbol{Z}_{\text{speech}}) + \mathbf{1}_{\text{audio}} \lambda \cdot \mathcal{L}_{\text{single}}(\boldsymbol{H}, \boldsymbol{Z}_{\text{audio}}),$$

where $\mathcal{L}_{\text{single}}$ is the single-domain masked prediction loss defined in Eqn. (1). The term $\mathbf{1}_{\text{audio}}$ is an indicator function that returns 1 if the input is general audio and 0 otherwise. $\lambda$ is a hyperparameter for balancing the contribution of the general-audio-specific loss.

### 3.2.3. TOKEN MIXING

Prior work (Chen et al., 2022; 2024b) performs a speech denoising task by only predicting the primary target while filtering out the secondary signal in a mixed training sample, which has been shown to improve the quality of speech representations. However, we argue that treating the secondary signals merely as "noise" to suppress may degrade the model's generality on complex sound scenes such as overlapped or noisy speech.

To address this, we introduce token mixing, a mechanism that dynamically constructs augmented training targets by stochastically combining the MVQ tokens from multiple sources based on the individual source energy for mixed audio samples. Let $\boldsymbol{w}$ be the original signal and $\boldsymbol{w}'$ be a randomly sampled signal, we mix them at their full length with a random delay to create an augmented training sample. The clean teacher targets $\boldsymbol{Z}$ and $\boldsymbol{Z}'$ of both signals are also mixed based on their signal power, forming an augmented

---

[2]If not, this can be achieved by interpolating the teacher representations if the frame rates differ.

[3]The effect of $\alpha$ is investigated in Appendix F.2

pre-training target $\hat{\boldsymbol{Z}}$:

$$
\hat{z}_{t,n} = \begin{cases} z_{t,n} & \text{with probability } 1 - \beta \\ z'_{t+\tau,n} & \text{with probability } \beta, \end{cases} \tag{4}
$$

where $\tau$ is the mixing delay (in frames) and $\beta \in [0, 1]$ is a scalar mixing coefficient derived from the signal power ($\mathcal{P}$) of the two mixed signals:

$$
\beta = \frac{\mathcal{P}(\boldsymbol{w}')}{\mathcal{P}(\boldsymbol{w}) + \mathcal{P}(\boldsymbol{w}')}. \tag{5}
$$

An illustration of the token mixing process is shown in Figure 2. By training against this mixed target, SPEAR jointly learns from both sources proportionally to their signal power, rather than strictly filtering out the secondary signal. The augmented pre-training target $\hat{\boldsymbol{Z}}$ will be used for loss computation following Eqn. (1).

## 4. Experimental Setup

**Dataset**   Pre-training is performed on both public unlabelled English speech datasets and general audio datasets, as shown in Table 1. Due to the limited amount of publicly available general audio datasets, we incorporate two music datasets, Music4all and MTG-Jamendo, to enrich the general audio data. The pre-training data used for different scales of SPEAR models are given in Table 2.

**Model Architecture**   As shown in (Shi et al., 2024a), modelling speech representations at different time resolutions improves speech SSL models. Therefore, Zipformer (Yao et al., 2024) is selected as the feature encoder in SPEAR due to its dynamic downsampling mechanism in the intermediate layers. The model receives 128-dimensional filter-bank features as input and produces frame-level representations at a 50 Hz frame rate. We pre-train three model sizes: Base (94M), Large (327M), and XLarge (600M). Detail regarding the encoder configurations is provided in Appendix B.1.

**Pre-training Configuration**   At the Base and Large scales, we pre-train both single-domain and dual-domain models. For the XLarge scale, we only train a dual-domain model with all the data. The pre-training data used for different scales of SPEAR models are given in Table 2. The training data groups are defined as follows: Speech-84k comprises Libriheavy, GigaSpeech, and VoxPopuli (en); Audio-13k includes all general audio datasets in Table 1; SA-97k combines Speech-84k and Audio-13k; and SA-197k additionally includes Yodas-granary. Token mixing is applied to 10% of all training samples, where the mixing signal is sampled from the same batch. Before mixing the waveforms, we sample an SNR between (-5, 5) and scale the second signal accordingly to match the SNR. Both signals are mixed across their full duration with a random offset. Other hyperparameters for pre-training are given in Appendix B.2.

*Table 1.* Pre-training corpora used in SPEAR.

| Domain | Dataset | Hours |
|---|---|---|
| Speech | Libriheavy (Kang et al., 2024) | 50k |
| | GigaSpeech (Chen et al., 2021) | 10k |
| | VoxPopuli (en) (Wang et al., 2021) | 24k |
| | Yodas-granary (en) (Koluguri et al., 2025) | 100k |
| Audio | AudioSet (Gemmeke et al., 2017) | 5k |
| | VGGsound (Chen et al., 2020) | 0.5k |
| | Freesound (Wu et al., 2023) | 2.8k |
| | Music4all (Santana et al., 2020) | 1k |
| | MTG-Jamendo (Bogdanov et al., 2019) | 3.8k |

*Table 2.* Pre-training configurations for different SPEAR settings.

| Model | Domain(s) | Data Groups | Total Hours |
|---|---|---|---|
| SPEAR$_s$-{Base, Large} | Speech | Speech-84k | ~84k |
| SPEAR$_a$-{Base, Large} | Audio | Audio-13k | ~13k |
| SPEAR$_{s+a}$-{Base, Large} | Speech & Audio | SA-97k | ~97k |
| SPEAR$_{s+a}$ XLarge | Speech & Audio | SA-197k | ~197k |

**Teacher Models and MVQ**   WavLM Large (Chen et al., 2022) and Dasheng 1.2B (Dinkel et al., 2024) are used to generate pre-training targets for speech and audio domains in the main paper, respectively. WavLM Large is a speech SSL model pre-trained on 94k hours of unlabelled English speech data, including Libri-light (Kahn et al., 2020), GigaSpeech, and Voxpopuli. It can be fairly compared with SPEAR models trained on Speech-84k since Libriheavy is the segmented version of Libri-light. The model generates a 1024-d feature at a 50 Hz frame rate. The speech MVQ quantiser with 16 codebooks is trained using the 21st layer representations of WavLM Large. Dasheng 1.2B is pre-trained with an MAE objective on an exceptionally large amount of speech and audio data of over 272k hours, roughly half of which is non-speech general audio. This model generates 1536-d features at 25 Hz. The audio MVQ quantiser is trained on the Dasheng final-layer representations with 8 codebooks. Ablation studies on different teacher models for pre-training target generation and quantiser configurations are presented in Appendix F.1.1 and Appendix F.5.

## 5. Experimental Results

To validate the effectiveness of the SPEAR framework, we assess its performance through both full fine-tuning and frozen representation evaluations on two major benchmarks for evaluating speech and audio representations: SUPERB (Yang et al., 2021; Tsai et al., 2022) and HEAR (Turian et al., 2022). Finally, ablation studies on core components of SPEAR are discussed in Section 5.4.

### 5.1. Downstream Fine-tuning

We evaluate the performance of SPEAR on two key downstream fine-tuning tasks: **automatic speech recognition**

*Table 3.* Fine-tuning results on ASR and AT. For ASR, WERs under "clean" and "other" denote the WERs on test-clean and test-other sets. $^{\triangle}$: Model fine-tuned from the public checkpoint. Best results in **bold**, 2nd best results underlined.

| Model | # Params | Pre-train data | LS-100 clean | LS-100 other | LS-960 clean | LS-960 other | AS-20k | AS-2M |
|---|---|---|---|---|---|---|---|---|
| *Speech SSL Models* | | | | | | | | |
| WavLM Base + $^{\triangle}$ | 95M | 94k | 4.0 | 8.4 | 2.9 | 5.4 | - | - |
| WavLM Large$^{\triangle}$ | 317M | 94k | 3.0 | 6.1 | 1.8 | 3.8 | - | - |
| Ours, SPEAR$_s$ Base | 94M | 84k | 3.0 | 5.7 | 1.9 | 4.0 | 26.9 | 43.6 |
| Ours, SPEAR$_s$ Large | 327M | 84k | 2.6 | 4.7 | 1.7 | 3.3 | 26.4 | 43.9 |
| *Audio SSL Models (AudioSet 5k hours)* | | | | | | | | |
| BEATs (Chen et al., 2023) | 90M | 5k | - | - | - | - | 38.9 | 48.6 |
| EAT (Chen et al., 2024a) | 88M | 5k | - | - | - | - | **40.2** | 48.6 |
| ATST Frame (Li et al., 2024) | 86M | 5k | - | - | - | - | 39.0 | 48.0 |
| Ours, SPEAR$_a$ Base | 94M | 5k | - | - | - | - | 39.0 | 49.3 |
| Ours, SPEAR$_a$ Large | 327M | 5k | - | - | - | - | 39.3 | 49.7 |
| Ours, SPEAR$_a$ Base | 94M | 13k | 11.2 | 23.0 | - | - | 39.2 | 49.3 |
| Ours, SPEAR$_a$ Large | 327M | 13k | 7.4 | 18.6 | - | - | 39.3 | 49.8 |
| *Speech & Audio SSL Models* | | | | | | | | |
| USAD Base$^{\triangle}$ | 94M | 126k | 4.9 | 11.4 | - | - | 35.7 | - |
| USAD Large$^{\triangle}$ | 330M | 126k | 4.0 | 7.7 | - | - | 37.4 | - |
| Ours, SPEAR$_{s+a}$ Base | 94M | 97k | 3.1 | 6.0 | 1.9 | 4.2 | 39.1 | 48.6 |
| Ours, SPEAR$_{s+a}$ Large | 327M | 97k | 2.6 | 4.8 | 1.7 | 3.4 | 39.2 | 49.6 |
| Ours, SPEAR$_{s+a}$ XLarge | 600M | 197k | **2.4** | **4.6** | **1.6** | 2.9 | 39.4 | **50.0** |

(ASR) for speech capabilities and **audio tagging** (AT) for general audio understanding capabilities. The results are presented in Table 3 and detailed fine-tuning configurations are shown in Appendix B.3.

**ASR** The ASR performance is evaluated on LibriSpeech (Panayotov et al., 2015), where the model is fine-tuned on the train-clean-100 subset (LS-100) or the full 960 hours (LS-960). The ASR decoder is a stateless RNN-T decoder (Ghodsi et al., 2020) with 3M parameters and a 500-unit BPE vocabulary (Sennrich et al., 2016). Performance is measured by the Word Error Rate (WER) on the test-clean and test-other sets of LibriSpeech, using beam search decoding with no external language model. Additional results of fine-tuning with a CTC (Graves et al., 2006) decoder are shown in Appendix C.

**Audio Tagging** To evaluate audio capabilities, our models are fine-tuned on AudioSet for AT following the procedure in Gong et al. (2021). We perform fine-tuning on the balanced subset (AS-20k) and the full dataset (AS-2M). A linear projection layer is added after the encoder to predict the probability of 527 sound event classes. Binary cross-entropy is used for optimisation, and the mean average precision (mAP) is measured on the AudioSet evaluation set.

**Results** The results presented in Table 3 demonstrate that SPEAR learns high-quality representations in both single-domain and dual-domain setups. In the speech domain, SPEAR$_s$ Base and Large achieve relative WER reductions of 25.9% and 13.2% on the test-other set in LS-960 compared to their WavLM counterparts with similar model size and pre-training data. In the audio domain, SPEAR$_a$ Base pre-trained on AudioSet achieves an mAP of 49.3 on AS-

2M, surpassing all other audio SSL models pre-trained on AudioSet. After increasing the model size, SPEAR$_a$ Large improves further to 49.7 on AS-2M.

Moreover, our dual-domain models successfully learn a unified representation space capable of handling both tasks with minimal performance loss comparable to their single-domain counterparts, and this performance gap diminishes as model capacity increases. Specifically, the WER for SPEAR$_{s+a}$ Large model on test-other is only 0.1 higher than the speech-domain specialist SPEAR$_s$ Large, while its mAP is only 0.2 lower than the SPEAR$_a$ Large. It should be noted that this versatility is particularly important, since all single-domain models show poor cross-domain transfer capability (see SPEAR$_s$ on AT and SPEAR$_a$ on ASR). Compared to USAD, SPEAR$_{s+a}$ consistently achieves better performance while using less pre-training data.

**5.2. SUPERB Evaluation**

**Setup** Experiments are carried out on SUPERB (Yang et al., 2021; Tsai et al., 2022), a benchmark for evaluating SSL representations on a wide range of speech processing tasks. We follow the standard SUPERB evaluation protocol, using a weighted sum of the frozen intermediate representations from the SSL models. For better readability, we group the SUPERB tasks into three categories: **Understanding**, **Paralinguistic**, and **Enhancement**. We select representative tasks within each category and report their results in Table 4. The primary comparison is made against WavLM Large, which is the current SOTA on SUPERB. We also compare our dual-domain models SPEAR$_{s+a}$ with USAD, another unified speech and audio model trained via multi-teacher feature matching. Further details and complete results on SUPERB can be found in Appendix D.

**Results** As can be seen from Table 4, SPEAR improves across the range of speech tasks, achieving notable gains across all three task categories compared to the current SOTA WavLM. Specifically, with the same model size and pre-training corpora, the speech domain SPEAR$_s$ Large model outperforms the current SOTA WavLM Large on nearly every task. The improvement in paralinguistic capabilities is particularly noteworthy: SPEAR$_s$ Large achieves a 16.7% relative reduction of equal-error-rate on speaker verification (SV) and a 1.5% absolute accuracy improvement on emotion recognition (ER). This suggests that masked prediction of fine-grained MVQ tokens in SPEAR helps the model learn richer paralinguistic information beyond lexical content alone than by using k-means tokens. An analysis of the feature subspaces learned through the MVQ tokens is conducted in Appendix F.4.2.

Our dual-domain SPEAR models maintain strong performance on SUPERB while being more versatile than speech-

*Table 4.* Results on SUPERB. Best results in **bold**, 2nd best underlined. Task metrics described in Appendix D. Results for USAD, BEATs, and EAT from (Chang et al., 2025).

| Model | # Param | Pre-train data | Understanding | | | | Paralinguistic | | | | Enhancement | |
|---|---|---|---|---|---|---|---|---|---|---|---|---|
| | | | PR↓ | ASR↓ | IC↑ | KS↑ | SID↑ | SV↓ | SD↓ | ER↑ | SE↑ | SS↑ |
| **Speech SSL models** | | | | | | | | | | | | |
| WavLM Base+ | 95M | 94k | 3.5 | 3.92 | 99.00 | 97.37 | 89.4 | 4.07 | 3.5 | 68.7 | 2.63 | 10.85 |
| WavLM Large | 317M | 94k | 3.1 | 3.44 | 99.31 | 97.86 | 95.5 | 3.77 | 3.2 | 70.6 | 2.70 | 11.19 |
| Ours, SPEAR$_s$ Base | 94M | 84k | 3.4 | 3.43 | 99.17 | 97.50 | 90.5 | 3.77 | 2.9 | 69.1 | 2.64 | 11.32 |
| Ours, SPEAR$_s$ Large | 327M | 84k | **2.6** | 3.24 | 99.44 | 97.88 | 95.6 | 3.14 | 2.3 | 72.1 | 2.72 | 11.67 |
| **Audio SSL models** | | | | | | | | | | | | |
| BEATs | 90M | 5k | 36.4 | 36.4 | 97.70 | 53.40 | 57.1 | - | 5.2 | 64.5 | - | - |
| EAT | 88M | 5k | 55.0 | 25.9 | 92.80 | 62.50 | 45.0 | - | 4.7 | 62.5 | - | - |
| Dasheng 1.2B | 1.2B | 272k | 14.3 | 13.8 | 98.13 | 97.73 | 92.4 | - | 3.8 | 68.7 | - | - |
| **Speech & Audio Models** | | | | | | | | | | | | |
| USAD Base | 94M | 126k | 5.1 | 7.70 | 98.30 | 97.10 | 88.6 | - | 4.2 | 68.0 | - | - |
| USAD Large | 330M | 126k | 4.0 | 6.50 | 98.40 | 97.10 | 91.2 | - | 3.9 | 68.4 | - | - |
| Ours, SPEAR$_{s+a}$ Base | 94M | 97k | 3.8 | 3.79 | 98.13 | 97.59 | 90.3 | 3.85 | 3.0 | 69.2 | 2.67 | 11.39 |
| Ours, SPEAR$_{s+a}$ Large | 327M | 97k | 3.1 | 3.39 | 99.40 | 97.92 | 95.2 | 3.30 | 2.3 | 71.7 | **2.73** | 11.60 |
| Ours, SPEAR$_{s+a}$ XLarge | 600M | 197k | 2.9 | **3.15** | **99.60** | **98.12** | **96.3** | **2.86** | **2.0** | **73.3** | **2.75** | **11.73** |

domain models. Despite a slight degradation in some understanding and paralinguistic tasks, SPEAR$_{s+a}$ Large outperforms its speech-only counterpart SPEAR$_s$ Large on keyword spotting (KS) and speech enhancement (SE), indicating a positive synergy from the dual-domain pre-training. SPEAR$_{s+a}$ also consistently outperforms USAD at comparable model sizes while using less training data. After scaling, SPEAR$_{s+a}$ XLarge pushes the performance boundary even further, establishing new SOTA on most tasks.

It is also evident that audio SSL models generally underperform on the SUPERB benchmark. Even with a very large pre-training corpus of 272k hours containing both speech and general audio data, Dasheng 1.2B consistently performs more poorly than our much smaller SPEAR$_{s+a}$ or SPEAR$_s$ Large, especially on understanding-based tasks (*e.g.*, ASR). This gap could be attributed to the MAE objective used by Dasheng, which tends to focus on low-level acoustic detail rather than high-level semantic structures necessary for speech understanding. In contrast, by leveraging fine-grained MVQ tokens extracted from domain-specific teachers, SPEAR allows the model to learn semantic structures while retaining acoustic details through masked-token prediction, leading to a comprehensive performance over both domains after pre-training on speech and audio data.

### 5.3. HEAR Evaluation

**Setup** To assess the general audio capabilities of our models, experiments are conducted on the HEAR benchmark (Turian et al., 2022), which evaluates audio representations across 19 diverse tasks. In contrast to SUPERB, HEAR focuses more on audio-related tasks requiring low-

level acoustic details. The final-layer representations are used for evaluation unless otherwise specified. For clarity, the average scores for each of the three task categories are reported: Environment, Speech and Music, along with the overall average score in Table 5. More information regarding the tasks in HEAR and detailed results on individual tasks are provided in Appendix E.

**Results** As shown in Table 5, all speech-domain models show limited overall performance on the audio-centric HEAR benchmark, highlighting the need to incorporate general audio data during pre-training. Nonetheless, our SPEAR$_s$ models yield a higher overall score than the much larger WavLM Large, indicating that the fine-grained MVQ tokens capture low-level acoustic detail despite being extracted from a speech SSL model.

Among all audio SSL models trained on AudioSet, SPEAR achieves the highest average score on HEAR. Increasing pre-training data to 13k hours further improves the audio-domain SPEAR models. Notably, SPEAR$_a$ Large achieves 83.58 on the environment category, surpassing the best performing audio model Dasheng 1.2B (83.2) pre-trained on 272k hours, with only a quarter of the model parameters and far less pre-training data. This highlights the potential of SPEAR for audio SSL at larger scales. However, the overall performance of SPEAR$_a$ Large on HEAR still trails that of Dasheng 1.2B, a gap we attribute to the significantly smaller scale of general audio training data and model size.

Our dual-domain SPEAR$_{s+a}$ models consistently outperform their single-domain counterparts, verifying the benefits of unified pre-training over speech and audio data. The SPEAR$_{s+a}$ Large achieves an average score of 79.26, sur-

*Table 5.* Results on the HEAR benchmark. The group-wise average score and the overall average score are reported. Rows with grey background: results obtained by using the concatenation of all layers' features. Best results in **bold**, 2nd best results underlined.

| Model | # Params | Pre-train Data | HEAR | | | |
|---|---|---|---|---|---|---|
| | | | Env | Speech | Music | Average |
| *Speech Models* | | | | | | |
| WavLM Base+ | 95M | 94k | 57.28 | 68.14 | 61.31 | 62.69 |
| WavLM Large | 317M | 94k | 72.86 | 72.69 | 65.77 | 69.65 |
| Ours, SPEAR$_s$ Base | 94M | 84k | 73.09 | 73.41 | 70.66 | 72.12 |
| Ours, SPEAR$_s$ Large | 327M | 84k | 72.74 | 74.80 | 71.68 | 72.96 |
| *Audio Models (AudioSet 5k hours)* | | | | | | |
| BEATs | 90M | 5k | 73.23 | 62.40 | 77.52 | 71.05 |
| Dasheng-Base | 86M | 5k | - | - | - | 70.43 |
| Dasheng 0.6B | 600M | 5k | - | - | - | 71.75 |
| Dasheng 1.2B | 1.2B | 5k | - | - | - | 74.87 |
| Ours, SPEAR$_a$ Base | 94M | 5k | 77.83 | 69.74 | 80.61 | 76.37 |
| Ours, SPEAR$_a$ Large | 327M | 5k | 78.16 | 72.94 | 81.80 | 78.08 |
| *Audio Models (others)* | | | | | | |
| Dasheng-Base | 86M | 272k | 80.18 | 72.48 | 84.00 | 79.31 |
| Dasheng 0.6B | 600M | 272k | 82.95 | 74.82 | 84.73 | 81.03 |
| Dasheng 1.2B | 1.2B | 272k | 83.20 | 75.72 | 84.86 | 81.44 |
| Ours, SPEAR$_a$ Base | 94M | 13k | 80.33 | 69.87 | 80.33 | 77.01 |
| Ours, SPEAR$_a$ Large | 327M | 13k | 83.58 | 72.70 | 81.85 | 79.18 |
| *Speech & Audio Models* | | | | | | |
| USAD Base | 94M | 126k | 80.67 | 73.72 | 79.31 | 77.75 |
| USAD Large | 330M | 126k | 81.97 | 74.48 | 81.7 | 79.36 |
| Ours, SPEAR$_{s+a}$ Base | 94M | 97k | 80.60 | 73.95 | 79.35 | 77.83 |
| Ours, SPEAR$_{s+a}$ Large | 327M | 97k | 81.10 | 76.47 | 80.42 | 79.26 |
| Ours, SPEAR$_{s+a}$ XLarge | 600M | 197k | 82.30 | 76.97 | 81.29 | 80.07 |
| Ours, SPEAR$_{s+a}$ Base | 94M | 97k | 82.55 | 77.45 | 83.61 | 81.32 |
| Ours, SPEAR$_{s+a}$ Large | 327M | 97k | 84.33 | 78.88 | 84.21 | 82.46 |
| Ours, SPEAR$_{s+a}$ XLarge | 600M | 197k | **84.85** | **79.75** | **85.43** | **83.41** |

passing both SPEAR$_s$ Large and SPEAR$_a$ Large, which suggests that joint speech–audio pre-training yields a more unified representation space. Compared to USAD Large, SPEAR$_{s+a}$ Large pre-trained on a smaller corpora achieves an absolute improvement of 3.12 on the average score under the same evaluation setup of using intermediate representations. This again highlights the strength of SPEAR as a unified SSL framework for learning generic speech and audio representations. Compared to the much larger Dasheng 1.2B, SPEAR$_{s+a}$ XLarge achieves stronger performance on speech-related tasks while trailing on environment and music tasks, which can be attributed to the imbalanced pre-training data composition (only 13k hours from the 197k hours used by SPEAR are general-audio data). However, it should be noted that SPEAR$_{s+a}$ XLarge is a more versatile model than Dasheng 1.2B, outperforming Dasheng 1.2B on SUPERB (see Table 4) by a large margin.

## 5.4. Ablation Studies

**Token Mixing** This ablation aims to investigate the effect of token mixing introduced in Section 3.2.3. By viewing WavLM-style denoising loss and our proposed token mixing mechanism effectively both as data augmentation methods, we compare three setups: without data augmentation, with WavLM-style loss (Chen et al., 2022), and with our proposed token mixing. The speech domain models are trained on LibriSpeech data using the Base scale with the aforementioned three configurations. Results are shown in Table 6.

As can be seen, both augmentations outperform the baseline without augmentations. While achieving similar WERs on ASR, token mixing outperforms WavLM-style denoising on speaker diarisation (SD), speech separation (SS), and speech enhancement (SE) by a large margin, three tasks requiring separation from different sources in the overlapped/noisy speech. This validates the effectiveness of token mixing when processing complex sound scenes.

*Table 6.* Comparison among no data augmentation, WavLM-style denoising loss, and our proposed token mixing mechanism.

| Setup | LS-100 | | SUPERB | | | | |
|---|---|---|---|---|---|---|---|
| | clean | other | PR ↓ | SD ↓ | SS ↑ | SID ↑ | SE ↑ |
| None | **3.1** | 7.1 | **3.3** | 4.23 | 10.52 | 87.5 | 2.60 |
| WavLM-style | **3.1** | **7.0** | 3.4 | 3.78 | 10.76 | **88.4** | 2.63 |
| Token Mixing | **3.1** | **7.0** | 3.4 | **3.01** | **11.17** | 88.3 | **2.65** |

**Dual-domain Training Strategy** As mentioned in Section 3.2.2, we adopt an asymmetrical pre-training loss for dual-domain pre-training. Here, we compare it with the other two strategies introduced in Section 3.2.2 using the dual-domain setup of Large size model on the SA-97k data (see Table 2). The models are evaluated after 100k training steps without token mixing augmentation. We compare the ASR and AT fine-tuning results, two SUPERB tasks, and the average score on HEAR, and the results are shown in Table 7. Among the three strategies, DISJOINT achieves the worst performance, indicating that treating both domains independently is harmful to unified representation learning. ASYMMETRICAL achieves the best overall performance, suggesting that learning speech MVQ tokens for audio data is a useful regularisation to bridge the domain mismatch. On the other hand, also computing pre-training loss against the audio MVQ tokens for speech data in the JOINT strategy leads to slightly worse overall performance. One possible reason is that audio MVQ tokens contain limited semantic information relevant to speech, degrading the pre-training performance, especially when the speech data is dominant.

*Table 7.* Results of three strategies for dual-domain pre-training.

| Strategy | LS-100 | | AS-20K↑ | SUPERB | | HEAR |
|---|---|---|---|---|---|---|
| | clean↓ | other↓ | | SID↑ | PR↓ | Avg↑ |
| JOINT | **2.9** | 5.9 | 36.9 | 90.6 | 3.34 | 78.7 |
| DISJOINT | 3.0 | **5.8** | **37.0** | 87.4 | 3.40 | 78.3 |
| ASYMMETRICAL | **2.9** | **5.8** | 36.9 | **90.7** | **3.12** | **79.0** |

**Other Ablations and Comparisons** Beyond the core results, we conducted extensive ablation studies to validate the design choices of the SPEAR framework:

- **Teacher Model Selection** (Appendix F.1.1): We experimented with other teacher models (e.g., HuBERT, ATST-frame) SPEAR consistently outperforms them

across domains, demonstrating that the student is not upper-bounded by the teacher and that stronger teachers yield stronger students.

- **Encoder Architectures** (Appendix F.3): Comparing Zipformer with a standard Transformer, we found the Zipformer offers better computational efficiency and stronger ASR fine-tuning performance.

- **Pre-training Target** (Appendix F.4): MVQ tokens consistently outperform standard k-means clusters, particularly on paralinguistic tasks (SID, ER). Feature subspace analysis confirms that even a single MVQ codebook retains rich speaker characteristics.

- **Pre-training Loss Weighting** (Appendix F.2): We found that balancing the prediction loss between masked and unmasked positions ($\alpha$) is crucial for stabilising the complex MVQ prediction task, while a tuned audio-loss weight ($\lambda$) optimally balances dual-domain performance.

- **MVQ Configurations** (Appendix F.5): Increasing the number of codebooks benefits speech modelling by capturing finer nuances, whereas a moderate number of codebooks is optimal for general audio.

- **Comparison with USAD** (Appendix G): Under a strictly controlled setup with identical teachers and scale, SPEAR outperforms USAD, confirming that discrete masked prediction effectively mitigates the cross-domain interference inherent in direct feature distillation.

## 6. Conclusions

In this work, we propose SPEAR, a unified SSL framework for learning unified and generic representations across both speech and general audio domains. SPEAR leverages MVQ to generate fine-grained discrete speech and audio tokens from domain-specific teachers, and performs masked-token prediction for representation learning over speech and audio data. Using an asymmetrical dual-domain pre-training objective, SPEAR learns a balanced and unified representation space. Furthermore, a novel token mixing mechanism is designed to enhance the performance for complex sound scenes. The downstream fine-tuning experiments, along with the evaluation of frozen representations on two major benchmarks for assessing speech and general-audio representations (SUPERB and HEAR), demonstrate the effectiveness of SPEAR in learning unified and generic speech and audio representations. Our dual-domain model with 600M parameters excels in both domains, making it one of the most powerful and versatile open-source SSL models for unified speech and audio understanding.

## 7. Limitations

While SPEAR achieves state-of-the-art performance in unifying speech and audio representations, we acknowledge a few limitations. First, the multi-teacher knowledge distillation framework introduces additional computational overhead during the data preparation and pre-training phases. This includes the prerequisite of training a MVQ quantiser for each teacher, as well as performing forward passes through the frozen teacher models to encode their continuous representations into discrete training targets.

Second, SPEAR is built upon the representations of high-quality, domain-specific teacher models. Consequently, the capacity of the chosen teachers inherently influences the overall performance of the student model, as with other knowledge distillation-based methods. In spite of this, our framework exhibits strong resilience to such constraints. As detailed in Appendix F.1.1, our ablation studies across different teacher models demonstrate that SPEAR consistently outperforms its respective teachers, indicating that the model learns robust and generalised representations rather than merely mimicking the teacher outputs.

Finally, SPEAR is predominantly focused on understanding-related tasks in the field of speech and general audio, rather than generative applications. Exploring how to seamlessly integrate both acoustic understanding and generative capabilities within a single, unified feature space remains an important direction for our future work.

## Impact Statement

This work presents a unified representation model for both speech domain and general audio domain. It could serve as a unified frontend for various fields including speech understanding, voice modelling, sound event detection. This comprehensive audio perception capability could make the work an essential component for building multi-modal AI systems.

We recognise that powerful audio representation models could be misused. The technology presented could serve as a foundation for applications we do not endorse, such as non-consensual speaker identification, mass surveillance, or the generation of synthetic audio for disinformation. Our goal in releasing these models is to enable transparency and accelerate positive academic innovation. We strongly condemn any application of our research to unethical ends.

## Acknowledgements

Xiaoyu Yang is supported by a Jardine Foundation Scholarship. The authors gratefully acknowledge this support.

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

# A. Multi-codebook Vector Quantisation

Here, we present more details regarding the MVQ quantiser to supplement Section 3.1.

## A.1. Encode and Decode

A Multi-codebook Vector quantisation (MVQ) module consists of $N$ codebooks, each containing $K$ codebook vectors. Given a $d$-dimensional representation $\boldsymbol{x} \in \mathbb{R}^d$, the MVQ quantiser $\mathcal{Q}$ encodes it to a sequence of integers (i.e, tokens) from a finite discrete value space $[0, \ldots, K-1]$.[4] The encoded integers are denoted as MVQ tokens, which can be used for reconstructing the original input through a $\mathrm{Decode}(\cdot)$ operation:

$$\boldsymbol{z} = \mathrm{Encode}(\boldsymbol{x}; \mathcal{Q}), \tag{6}$$

$$\hat{\boldsymbol{x}} = \mathrm{Decode}(\boldsymbol{z}; \mathcal{Q}). \tag{7}$$

The MVQ quantiser performs a mapping $f : \mathbb{R}^d \to \mathbb{C}^N$, where $\mathbb{C}$ denotes a fixed-sized discrete value space $\{0, \ldots, K-1\}$. $\boldsymbol{z} = \{z_1, \ldots, z_N\} \in \mathbb{C}^N$ is the MVQ tokens with $z_n \in \{0, \ldots, K-1\}$ and $\hat{\boldsymbol{x}}$ is the reconstructed input. The reconstruction operation follows a direct-sum scheme, where one codebook vector is selected from each codebook, resulting in a summation over $N$ codebook vectors:

$$\hat{x} = \sum_{n=1}^{N} \boldsymbol{C}_{z_n}^n, \tag{8}$$

where $\boldsymbol{C}_{z_n}^n \in \mathbb{R}^d$ is the $z_n$-th entry code vector in the $n$-th codebook. Each codebook $\boldsymbol{C}^n = \{\boldsymbol{c}_0^n, \ldots, \boldsymbol{c}_{K-1}^n\}$ is a matrix consisting of $K$ code vectors. As can be seen, $z_n$ denotes the encoded index of the code vector in the $n$-th codebook, i.e., which code vector to choose from the $n$-th codebook for reconstruction.

The encoding process aims to find $\boldsymbol{z}$ that leads to the lowest reconstruction error: $E[||\hat{\boldsymbol{x}} - \boldsymbol{x}||_2^2]$. Naively enumerating all combinations of $z_n$ is impractical, so a heuristic encoding algorithm is utilised to reduce the search space while maintaining a relatively low reconstruction error. The MVQ quantiser employs $N$ neural classifiers $\mathcal{G}_n$ to first generate an initial estimation of the encoded index for each codebook, denoted as $\boldsymbol{z}_{\mathrm{init}}$, and iteratively refines $\boldsymbol{z}_{\mathrm{init}}$ for a fixed number of steps, e.g., 5. The mechanism of the refinement algorithm is out of the scope of our work, and we direct readers to the original MVQ paper (Guo et al., 2023) for more details.

## A.2. MVQ Training

The trainable parameters in the MVQ quantiser are the codebooks $\boldsymbol{C}^n$ and $N$ neural classifiers $\mathcal{G}_n$. For each input float vector $\boldsymbol{x}$ and its encoding $\boldsymbol{z} = \mathrm{Encode}(\boldsymbol{x})$, the training loss for MVQ is formulated as follows:

$$\mathcal{L} = \mathcal{L}_{\mathrm{residual}} + \mathcal{L}_{\mathrm{prediction}} + \gamma \mathcal{L}_{\mathrm{reg}} \tag{9}$$

$$= ||\boldsymbol{x} - \mathrm{Decode}(\boldsymbol{z}; \mathcal{Q})||_2^2 + \sum_{n=1}^{N} -\log \mathcal{G}_n(\boldsymbol{x})_{z_n} + \gamma \mathcal{L}_{\mathrm{reg}}, \tag{10}$$

where $\mathcal{G}_n(\boldsymbol{x})_{z_n}$ is the predicted probability of choosing $z_n$. The first term $\mathcal{L}_{\mathrm{residual}}$ is the L2-squared reconstruction loss and optimises the code vectors. The second term $\mathcal{L}_{\mathrm{prediction}}$ encourages the neural classifiers to select the encoded indexes $\boldsymbol{z}$ obtained through the refinement algorithm. By doing so, the initial estimate $\boldsymbol{z}_{\mathrm{init}}$ predicted by $\mathcal{G}_n$ is expected to be close to the actual encodings $\boldsymbol{z}$, most likely with a lower reconstruction error. The last term $\mathcal{L}_{\mathrm{reg}}$ is an auxiliary regularisation loss to encourage a balanced code usage within each codebook, and $\gamma$ is the scale for this auxiliary loss.

# B. Model Specification and Training Settings

## B.1. Model Specification

The model specifications of Base, Large, and XLarge variants of SPEAR are presented in Table 8. The model configuration is determined by the configuration of the Zipformer (Yao et al., 2024) encoder, which adopts a stack-wise design, with each

---

[4]For storage efficiency, we always use $K = 256$ since the indices can be stored with uint8 format. However, it is theoretically possible to increase $K$ to a bigger number.

stack consisting of multiple layers operating at a specific downsampling factor. The Zipformer Encoder is characterised by the following attributes:

- **Model Dimension**: the dimensionality of the output representations.

- **Feedforward Dimension**: the dimensionality of the feedforward module.

- **Attention Heads**: the number of attention heads.

- **Encoder Layers**: the number of Zipformer layers per stack.

- **Downsampling Ratio**: the relative temporal downsampling factor to the input representations (i.e., 100 Hz filterbank features).

- **CNN Kernel Size**: the kernel size of the convolutional module in each layer.

*Table 8.* The configurations of the Zipformer encoder in different versions of SPEAR.

|                      | Base           | Large          | XLarge         |
|----------------------|----------------|----------------|----------------|
| Number of parameters | 94M            | 327M           | 600M           |
| Model dimension      | 512            | 1024           | 1280           |
| Feedforward dimension| 1536           | 3072           | 3840           |
| Attention heads      | 8              | 8              | 8              |
| Encoder layers       | 1,2,3,3,1,1,1  | 1,2,2,3,1,1,1  | 1,2,3,4,1,1,1  |
| Downsampling ratio   |                | 1,2,4,8,4,2,1  |                |
| CNN kernel size      |                | 31,31,15,15,15,31,31 |          |

### B.2. Pre-training Resources and Configurations

The training hyperparameters and the required computing resources for training SPEAR are presented in Table 9. All models in SPEAR are trained using the NVIDIA A800 (80GB) GPUs. As shown in WavLM (Chen et al., 2022), applying data augmentations to the input audio can improve the pre-training performance. Therefore, we apply both in-batch utterance mixing and noise mixing during training. MUSAN (Snyder et al., 2015) is used as the noise dataset. The optimiser and scheduler settings follow (Yao et al., 2024), where the ScaledAdam optimiser and Eden scheduler are used. During pre-training, the Token Mixing is applied to 10% of the data, where the secondary signal is sampled from the same batch. During dual-domain training, we mix either speech or audio data into the speech training sample to simulate overlapped or noisy speech and perform token mixing using the MVQ tokens generated by the speech SSL teacher.

### B.3. Fine-tuning Configurations

The fine-tuning configurations for the downstream ASR tasks and AT tasks presented in Section 5.1 are shown in Table 10. We used Pruned RNN-T (Kuang et al., 2022), a memory-efficient variant of RNN-T for optimisation. An asynchronous learning rate policy is adopted during fine-tuning by setting a smaller learning rate for the pre-trained encoder parameters. In ASR experiments, a 3-fold speed perturbation is applied to the training data. In AS-2M, we follow prior work (Gong et al., 2021) to adopt a weighted sampler to cope with the imbalanced label distribution in the full AudioSet. Mixup (Zhang et al., 2018) with a probability of 0.5 is used in AT fine-tuning.

## C. LibriSpeech Fine-tuning Experiments

To evaluate the adaptation capability of SPEAR under limited supervision, we fine-tune the models on 10h and 100h subsets of the LibriSpeech corpus. Following prior work (Hsu et al., 2021; Chen et al., 2022), we use the same CTC decoder with graphemes as modelling units and the same decoding process for fair comparison. The CTC vocabulary consists of the 26 English letters, a space, an apostrophe, and a blank symbol. Decoding with an external language model is performed using the wav2letter++ beam search decoder (Pratap et al., 2019), formulated as:

$$\log p_{\text{CTC}}(\boldsymbol{y} \mid \boldsymbol{x}) + w_1 \log p_{\text{LM}}(\boldsymbol{y}) + w_2 \, |\boldsymbol{y}|, \tag{11}$$

*Table 9.* Hyperparameters and computing resources required for pre-training. Batch size denotes the total duration of speech (audio) in seconds. Approximate total GPU hours (not elapsed time) also reported.

| | Speech Pre-train | | Audio Pre-train | | Speech & Audio Pre-train | | |
| --- | --- | --- | --- | --- | --- | --- | --- |
| | Base | Large | Base | Large | Base | Large | XLarge |
| **Hyperparameters** | | | | | | | |
| Learning rate | | | | 0.045 | | | |
| Total steps | 400k | 500k | 250k | 250k | 400k | 500k | 500k |
| Batch size | 4.8k | 4.8k | 4.8k | 4.8k | 6.4k | 6.4k | 6.4k |
| Utterance mix prob | 0.1 | 0.1 | - | - | - | - | - |
| Noise mix prob | 0.1 | 0.2 | 0.5 | 0.5 | 0.5 | 0.5 | 0.5 |
| $\alpha$ (see Equation 1) | | | | 0.5 | | | |
| $\lambda$ (see Equation 4) | - | - | - | - | 0.1 | 0.1 | 0.1 |
| **Computing Resources** | | | | | | | |
| Num GPUs | 8 | 8 | 8 | 8 | 8 | 16 | 32 |
| GPU hours (approx.) | 460 | 900 | 290 | 560 | 660 | 2,000 | 3,800 |

*Table 10.* Fine-tuning configurations. "Encoder LR scale" denotes the relative ratio of the encoder learning rate. Batch size is measured in seconds.

| | ASR | | AT | |
| --- | --- | --- | --- | --- |
| | LS-100 | LS-960 | AS-20k | AS-2M |
| Learning rate | | 0.045 | | |
| Encoder LR scale | 0.1 | 0.1 | 0.2 | 0.1 |
| Num epochs | 90 | 90 | 20 | 40 |
| Batch size | 2000 | 4800 | 2000 | 4000 |
| MUSAN (Snyder et al., 2015) | | | ✓ | |
| SpecAugment (Park et al., 2019) | | | ✓ | |
| Weighted sampling (Gong et al., 2021) | - | - | ✓ | ✓ |
| MixUp (Zhang et al., 2018) | - | - | 0.5 | 0.5 |

where $x$ is the input audio, $y$ is the predicted text sequence, $|y|$ denotes its length, and $w_1, w_2$ are the language model and word score coefficients, respectively.

**Result Interpretation** As shown in Table 11, the speech-domain $\text{SPEAR}_s$ models consistently outperform their WavLM counterparts under both Base and Large scales, regardless of decoding with an external 4-gram language model. Our $\text{SPEAR}_s$ Large yields the lowest WERs on both LS-10 and LS-100 setups, implying that the MVQ tokens used during pre-training transfer rich semantic information to the student model. Interestingly, the speech-domain $\text{SPEAR}_s$ models slightly outperform their dual-domain $\text{SPEAR}_{s+a}$ counterparts in this CTC setting, especially when supervision is scarce. We hypothesise this stems from the nature of the representation space: a unified representation space for speech and general audio is inherently more complex than one specialised for speech. Consequently, adapting the unified representation space for the ASR task becomes more challenging with insufficient supervised data, particularly when using a simple, letter-based CTC decoder.

## D. SUPERB Evaluation

In this section, we provide more detail about the SUPERB benchmark (Yang et al., 2021; Tsai et al., 2022) as a supplement to Section 5.2 and present the complete results on the SUPERB benchmark. A summary of the SUPERB tasks is shown in Table 12. Complete results of SPEAR models along with other existing speech SSL models are shown in Table 13 and Table 14.

*Table 11.* LibriSpeech fine-tuning results with limited supervised data. Best results in **bold**, and second-best results are underlined in each section.

| Model | # Params | LM | LS-100 | | LS-10 | |
|---|---|---|---|---|---|---|
| | | | test-clean | test-other | test-clean | test-other |
| WavLM Base (Chen et al., 2022) | 95M | None | 5.7 | 12.0 | 9.8 | 16.0 |
| WavLM Base+ (Chen et al., 2022) | 95M | None | 4.6 | 10.1 | 9.0 | 14.7 |
| Ours, SPEAR$_s$ Base | 94M | None | 3.1 | 6.0 | 5.2 | 8.2 |
| Ours, SPEAR$_{s+a}$ Base | 94M | None | 3.3 | 6.6 | 5.6 | 9.2 |
| Ours, SPEAR$_s$ Large | 327M | None | **2.6** | **4.8** | **4.6** | **6.9** |
| Ours, SPEAR$_{s+a}$ Large | 327M | None | **2.6** | 4.9 | 4.9 | 7.3 |
| Ours, SPEAR$_{s+a}$ XLarge | 600M | None | **2.6** | 4.9 | 4.8 | **6.9** |
| HuBERT Base (Hsu et al., 2021) | 95M | 4-gram | 3.4 | 8.1 | 4.3 | 9.4 |
| WavLM Base (Chen et al., 2022) | 95M | 4-gram | 3.4 | 7.7 | 4.3 | 9.2 |
| WavLM Base+ (Chen et al., 2022) | 95M | 4-gram | 2.9 | 6.8 | 4.2 | 8.8 |
| WavLM Large (Chen et al., 2022) | 317M | 4-gram | **2.3** | 4.6 | **2.9** | 5.5 |
| Ours, SPEAR$_s$ Base | 94M | 4-gram | 2.4 | 5.0 | 3.2 | 6.0 |
| Ours, SPEAR$_{s+a}$ Base | 94M | 4-gram | 2.7 | 5.3 | 3.6 | 6.9 |
| Ours, SPEAR$_s$ Large | 327M | 4-gram | **2.3** | **4.2** | **2.9** | **5.1** |
| Ours, SPEAR$_{s+a}$ Large | 327M | 4-gram | 2.4 | 4.4 | 3.1 | 5.6 |
| Ours, SPEAR$_{s+a}$ XLarge | 600M | 4-gram | **2.3** | 4.3 | **2.9** | 5.2 |

*Table 12.* Detailed task information in SUPERB.

| Task Name | Metric(s) |
|---|---|
| Speaker Identification (SID) | Accuracy |
| Speaker Verification (SV) | Equal Error Rate (EER) |
| Speaker Diarization (SD) | Diarization error rate (DER) |
| Emotion Recognition (ER) | Accuracy |
| Phoneme Recognition (PR) | Phone Error Rate (PER) |
| Automatic Speech Recognition (ASR) | Word Error Rate (WER) |
| Out-of-domain ASR (ar/es/zh) | Word (character) Error Rate |
| Keyword Spotting (KS) | Accuracy |
| Query by Example Spoken Term Detection (QbE) | Maximum Term Weighted Value (MTWV) |
| Speech Translation (ST) | BLEU |
| Intent Classification (IC) | Accuracy |
| Slot Filling (SF) | F1, Character Error Rate (CER) |
| Speech Enhancement (SE) | Perceptual Evaluation of Speech Quality (PESQ) Short-Time Objective Intelligibility (STOI) |
| Speech Separation (SS) | Scale-invariant Signal-to-distortion Ratio improvement (SI-SDRi) |
| Voice Conversion (VC) | MCD (Mel Cepstral Distortion), WER, EER |

**Result Interpretation** As shown in Table 13 and Table 14, our speech-domain model SPEAR$_s$ demonstrates very high performance on understanding, paralinguistics, and enhancement tasks on SUPERB. Our speech-domain model SPEAR$_s$ Large achieves better performance on 12 out of 15 tasks on SUPERB (except for ST, QbE, and VC) compared to WavLM Large, the previous state-of-the-art model on SUPERB, with the same pre-training corpora and similar model size. This

*Table 13.* Full results of understanding tasks on the SUPERB benchmark. Best results in **bold**, the 2nd best results are underlined.

| Model | # Params | Pre-train Data | Understanding | | | | | | | | |
|---|---|---|---|---|---|---|---|---|---|---|---|
| | | | PR | ASR | OOD-ASR | KS | QbE | ST | IC | SF | |
| | | | PER ↓ | WER ↓ | WER ↓ | Acc ↑ | MTWV ↑ | BLEU ↑ | Acc ↑ | F1 ↑ | CER ↑ |
| FBANK | 0 | - | 82.01 | 23.18 | 63.58 | 8.63 | 0.0058 | 2.32 | 9.10 | 69.64 | 52.94 |
| ***Existing Speech SSL models*** | | | | | | | | | | | |
| WavLM Base+ (Chen et al., 2022) | 95M | 94k | 3.92 | 5.59 | 38.32 | 97.37 | **0.0988** | 24.25 | 99.00 | 90.58 | 21.20 |
| wav2vec 2.0 Large (Baevski et al., 2020) | 317M | 60k | 4.25 | 3.75 | 44.89 | 96.66 | 0.0480 | 12.48 | 95.28 | 87.11 | 27.31 |
| HuBERT Large (Hsu et al., 2021) | 317M | 60k | 3.53 | 3.62 | 44.08 | 95.29 | 0.0353 | 20.10 | 98.76 | 89.81 | 21.76 |
| WavLM Large (Chen et al., 2022) | 317M | 94k | 3.06 | 3.44 | 32.27 | 97.86 | 0.0886 | 26.57 | 99.31 | 92.21 | 18.36 |
| ***Ours, Speech SSL models*** | | | | | | | | | | | |
| SPEAR$_s$ Base | 94M | 84k | 3.34 | 3.43 | 34.20 | 97.50 | 0.0772 | 24.37 | 99.17 | 90.96 | 19.22 |
| SPEAR$_s$ Large | 327M | 84k | **2.56** | 3.24 | 31.66 | 97.88 | 0.0768 | 26.20 | 99.44 | 92.25 | 17.86 |
| ***Ours, Speech & Audio SSL models*** | | | | | | | | | | | |
| SPEAR$_{s+a}$ Base | 94M | 97k | 3.84 | 3.79 | 35.37 | 97.58 | 0.0801 | 24.07 | 98.05 | 90.54 | 20.14 |
| SPEAR$_{s+a}$ Large | 327M | 97k | 3.08 | 3.39 | 31.20 | 97.92 | 0.0712 | 25.64 | 99.40 | 92.07 | 18.04 |
| SPEAR$_{s+a}$ XLarge | 600M | 197k | 2.92 | **3.15** | **30.51** | **98.12** | 0.0745 | **26.66** | **99.60** | **92.86** | **17.23** |

*Table 14.* Full results of paralinguistics, enhancement, and Generation tasks on the SUPERB benchmark. Best results in **bold**, the 2nd best results are underlined.

| Model | # Params | Pre-train Data | Paralinguistics | | | | Enhancement | | | Generation | | |
|---|---|---|---|---|---|---|---|---|---|---|---|---|
| | | | SID | SV | SD | ER | SE | | SS | VC | | |
| | | | Acc ↑ | EER ↓ | DER ↓ | Acc ↑ | PESQ ↑ | STOI ↑ | SI-SDRi ↑ | MCD ↓ | WER ↓ | SV ↑ |
| FBANK | 0 | - | 0 | 9.56 | 10.05 | 35.39 | 2.55 | 93.6 | 9.23 | 8.47 | 38.3 | 77.25 |
| ***Existing Speech SSL models*** | | | | | | | | | | | | |
| WavLM Base+ (Chen et al., 2022) | 95M | 94k | 89.42 | 4.07 | 3.50 | 68.65 | 2.63 | 94.3 | 10.85 | 7.40 | 8.1 | 99.00 |
| wav2vec 2.0 Large (Baevski et al., 2020) | 317M | 60k | 86.14 | 5.65 | 5.62 | 65.64 | 2.52 | 94.0 | 10.02 | 7.63 | 15.8 | 97.25 |
| HuBERT Large (Hsu et al., 2021) | 317M | 60k | 90.33 | 5.98 | 5.75 | 67.62 | 2.64 | 94.2 | 10.45 | **7.22** | **9.0** | **99.25** |
| WavLM Large (Chen et al., 2022) | 317M | 94k | 95.49 | 3.77 | 3.24 | 70.62 | 2.70 | 94.5 | 11.19 | 7.30 | 9.9 | 99.00 |
| ***Ours, Speech SSL models*** | | | | | | | | | | | | |
| SPEAR$_s$ Base | 94M | 84k | 90.5 | 3.77 | 2.93 | 69.14 | 2.64 | 94.3 | 11.32 | 7.40 | 10.1 | 99.00 |
| SPEAR$_s$ Large | 327M | 84k | 95.55 | 3.14 | 2.3 | 72.10 | 2.72 | 94.5 | 11.67 | 7.33 | 10.4 | 99.00 |
| ***Ours, Speech & Audio SSL models*** | | | | | | | | | | | | |
| SPEAR$_{s+a}$ Base | 94M | 97k | 90.26 | 3.85 | 3.0 | 69.24 | 2.67 | 94.5 | 11.39 | 7.34 | 10.2 | 99.00 |
| SPEAR$_{s+a}$ Large | 327M | 97k | 95.19 | 3.30 | 2.31 | 71.66 | **2.73** | **94.6** | 11.60 | 7.42 | 10.7 | 99.00 |
| SPEAR$_{s+a}$ XLarge | 600M | 197k | **96.34** | **2.86** | **1.99** | **73.29** | **2.75** | **94.6** | **11.73** | 7.44 | 10.9 | 99.00 |

again suggests that the performance of the SPEAR framework is not constrained by the teacher model, since SPEAR$_{s+a}$ Large is pre-trained with fine-grained targets generated by WavLM Large. It has also been observed that performing dual-domain training leads to slight performance degradation on understanding and paralinguistic tasks compared to the speech-only models. However, improvement on KWS and enhancement tasks is also observed, suggesting a positive task synergy in these tasks. Finally, our largest dual-domain model, SPEAR$_{s+a}$ XLarge, improves upon SPEAR$_{s+a}$ Large, and further improves the best results of 12 SUPERB tasks, confirming that the SPEAR framework scales effectively with both model and data size.

It is worth noting that our results on the Voice Conversion (VC) task are not directly comparable to previous SSL models due to a change in the VC recipe within the SUPERB codebase, as pointed out by Shi et al. (2024a). Within our own experiments, we observe that our Large and XLarge variants underperform the Base model on VC. This is potentially due to overfitting on the small training dataset, an issue also observed in MR-HUBERT (Shi et al., 2024a). We leave the investigation of using our models for generation tasks as an important direction for future work, since a unified capability for both understanding

and generation is highly desirable.

# E. HEAR Evaluation

Here, we provide further details on Holistic Evaluation of Audio Representations (HEAR) (Turian et al., 2022), a benchmark for evaluating audio representations as a supplement to Section 5.3. HEAR encompasses 19 tasks, which can be categorised into 3 groups: environment, speech, and music. Following prior work (Anton et al., 2023; Dinkel et al., 2024), we discard the Beehive task due to its overly long utterances and small sample size, leading to inconsistent results. The tasks can also be divided into frame-level tasks and clip-level tasks. The detailed task information is shown in Table 15 and the complete results on the HEAR benchmark are shown in Table 16.

*Table 15.* Individual task information in HEAR. $^*$: frame-level task. Otherwise, clip-level task.

| Task Name | Group | Description | Metric |
|---|---|---|---|
| Beijing Opera Percussion (BJ) | Music | Classification of 6 Beijing Opera percussion instruments | Accuracy |
| CREMA-D (CD) | Speech | Speech emotion recognition | Accuracy |
| DCASE 2016 Task2 (D16)$^*$ | Environment | Office sound event detection in synthesized scenes | Onset FMS |
| ESC-50 (ESC) | Environment | Environmental sound classification | Accuracy |
| FSD50K (FSD) | Environment | Broad-domain audio multi-labeling | mAP |
| Gunshot Triangulation (Gun) | Environment | Identify location of microphone recording a gunshot | Accuracy |
| GTZAN Genre (GZ-Gen) | Music | Music genre classification. | Accuracy |
| GTZAN Music Speech (GZ-MS) | Music | Classification of audio into music or speech. | Accuracy |
| LibriCount (LC) | Speech | Multiclass speaker count identification. | Accuracy |
| MAESTRO 5h (MST)$^*$ | Music | Music transcription | Onset FMS |
| Mridingham Stroke (Mri-S) | Music | Non-Western pitched percussion, classification of stroke | Accuracy |
| Mridingham Tonic (Mri-T) | Music | Non-Western pitched percussion, classification of tonic | Accuracy |
| NSynth Pitch, 5h (NS-5) | Music | Pitch classification of synthesized sounds. | Pitch Acc |
| NSynth Pitch, 50h (NS-50) | Music | Pitch classification of synthesized sounds. | Pitch Acc |
| Speech Commands (v2), 5h (SC-5) | Speech | Spoken commands classification. | Accuracy |
| Speech Commands (v2), full (SC-F) | Speech | Spoken commands classification. | Accuracy |
| Vocal Imitations (VI) | Speech | Classification of vocal imitation to type of sound imitated | mAP |
| VoxLingua107 Top 10 (VL) | Speech | Spoken language identification. | Accuracy |

**Result Interpretation**   Our audio-domain models demonstrate strong performance on the environment-related tasks. Specifically, SPEAR$_{s+a}$ Large outperforms its teacher model, Dasheng 1.2B, on D16, ESC, and FSD, three well-known tasks for environmental audio understanding. It should be noted that this is achieved under the premise that Dasheng 1.2B is a much bigger model trained on over 20 times more data. The performance of SPEAR$_a$ Large is lower on speech and audio tasks compared to Dasheng 1.2B, due to the limited amount of general audio data in our setup. However, we anticipate SPEAR to outperform Dasheng 1.2B given a similar amount of general audio data for pre-training, which we leave as an important direction for future work.

By performing unified speech and audio pre-training that incorporates more speech data, the dual-domain model SPEAR$_{s+a}$ yields a notable improvement on speech-related tasks over the audio-domain model SPEAR$_a$, as evidenced by tasks such as CD, SF-5, VI, and VL. We also observe a sharp increase for MST, a music transcription task, from SPEAR$_a$ Large with 23.6 to SPEAR$_{s+a}$ Large with 26.9. This suggests that the joint pre-training on speech and audio data enhances the model's capability of performing fine-grained music tasks. Despite achieving a better overall score on HEAR, we do notice that the dual-domain model suffers from performance degradation in some environments and music-related tasks. This further motivates us to use a more balanced dataset containing more general audio data in future work.

Finally, our largest dual-domain model SPEAR$_{s+a}$ XLarge achieves further performance improvement over SPEAR$_{s+a}$ Large, demonstrating that scaling data and model size is effective for SPEAR.

# F. Ablation Studies

Ablation studies on the following components are performed to provide an in-depth understanding of SPEAR framework:

*Table 16.* Results on the HEAR benchmark. The last column is the average performance across all tasks. All results are the higher the better. Rows in grey: evaluated by concatenating the intermediate layers. Best results in **bold**, the 2nd best results are underlined.

| Model | # Params | BJ | CD | D16 | ESC | FSD | GZ-Gen | GZ-MS | Gun | LC | MST | Mri-S | Mri-T | NS-50 | NS-5 | SC-5 | SC-F | VI | VL | Avg |
|---|---|---|---|---|---|---|---|---|---|---|---|---|---|---|---|---|---|---|---|---|
| **Speech Model** | | | | | | | | | | | | | | | | | | | | |
| WavLM Base+ | 96M | 87.3 | 68.7 | 49.9 | 60.1 | 32.8 | 75.0 | 98.5 | 86.3 | 62.5 | 4.3 | 89.8 | 78.9 | 35.1 | 21.6 | 94.4 | 95.1 | 14.2 | 74.0 | 62.7 |
| HuBERT Large | 317M | 92.4 | 74.5 | 44.0 | 64.5 | 35.8 | 74.7 | 92.8 | 94.4 | 64.3 | 3.1 | 95.1 | 85.9 | 39.4 | 19.8 | 91.5 | 92.8 | 17.5 | 73.3 | 64.2 |
| WavLM Large | 317M | 91.5 | 75.5 | 85.1 | 68.6 | 40.1 | 80.0 | 94.4 | 97.6 | 70.3 | 8.8 | 96.0 | 88.8 | 43.8 | 23.0 | 94.8 | 96.1 | 19.5 | 79.9 | 69.7 |
| $\text{SPEAR}_s$ Base | 94M | 94.5 | 79.5 | 92.0 | 74.8 | 43.4 | 83.9 | 96.0 | 82.1 | 66.5 | 6.3 | 95.6 | 90.5 | 61.7 | 36.8 | 95.9 | 96.4 | 20.3 | 81.9 | 72.1 |
| $\text{SPEAR}_s$ Large | 327M | 91.5 | 80.9 | 84.2 | 73.9 | 43.3 | 82.5 | 96.1 | 89.6 | 71.1 | 8.6 | 95.8 | 89.7 | 67.7 | 41.6 | 95.5 | 95.7 | 20.7 | 85.0 | 73.0 |
| **Audio Model** | | | | | | | | | | | | | | | | | | | | |
| BEATs | 90M | 95.8 | 68.1 | 43.0 | 81.9 | 51.4 | 87.0 | 98.5 | 90.5 | 74.6 | 0.0 | 96.1 | 96.0 | 82.0 | 68.6 | 88.2 | 91.5 | 13.5 | 43.8 | 69.3 |
| ATST-Frame | 86M | 95.8 | 76.7 | 95.7 | 89.0 | 55.7 | 88.3 | 100.0 | 94.3 | 78.1 | 24.4 | 97.5 | 94.1 | - | 68.6 | 92.6 | 95.1 | 22.3 | 66.9 | - |
| Dasheng base | 86M | 93.6 | 78.7 | 93.9 | 82.9 | 51.0 | 89.2 | 99.2 | 92.9 | 76.6 | **43.9** | 96.1 | 94.9 | 83.3 | 71.8 | 95.9 | 97.1 | 16.7 | 69.9 | 79.3 |
| Dasheng 0.6B | 600M | 94.9 | 81.2 | 94.4 | 85.9 | 53.9 | 88.6 | 97.6 | 97.6 | 80.7 | 43.5 | 96.6 | 96.2 | 85.8 | 74.6 | 97.0 | 97.5 | 17.8 | 74.7 | 81.0 |
| Dasheng 1.2B | 1.2B | **96.2** | 81.6 | 94.2 | 85.3 | 54.2 | 88.8 | 97.7 | **99.1** | 79.6 | 43.3 | 96.8 | 96.1 | 85.6 | 74.4 | 97.1 | 97.9 | 19.4 | 78.7 | 81.4 |
| $\text{SPEAR}_a$, Base | 94M | 93.6 | 77.2 | 93.6 | 85.5 | 52.9 | 90.1 | 92.2 | 89.3 | 77.2 | 22.8 | 96.7 | 96.5 | 83.7 | 70.0 | 93.8 | 95.1 | 18.8 | 57.1 | 77.0 |
| $\text{SPEAR}_a$, Large | 327M | 94.9 | 79.8 | 94.5 | 86.6 | 54.4 | 89.1 | 96.8 | 98.8 | 79.6 | 23.6 | 96.9 | 96.3 | 86.4 | 70.8 | 95.3 | 96.3 | 19.0 | 66.2 | 79.2 |
| $\text{SPEAR}_a$, Base | 94M | **96.2** | 79.0 | 95.4 | 87.4 | 55.3 | 89.4 | **100.0** | 96.4 | 81.6 | 23.8 | 97.0 | 97.5 | 87.5 | 74.8 | 94.9 | 95.9 | 19.9 | 60.7 | 79.6 |
| $\text{SPEAR}_a$, Large | 327M | 95.8 | 79.9 | **96.5** | **89.6** | **57.4** | 90.7 | 98.5 | 96.4 | **82.4** | 25.6 | **97.4** | **98.1** | **89.6** | **81.4** | 95.8 | 96.9 | 20.5 | 62.7 | 80.8 |
| **Speech + Audio Model** | | | | | | | | | | | | | | | | | | | | |
| USAD Base | 94M | 95.8 | 80.0 | 93.6 | 82.2 | 52.2 | 94.0 | **100.0** | 86.3 | 78.7 | 26.7 | 97.3 | 95.7 | 81.6 | 57.0 | 96.6 | 97.6 | 19.5 | 76.0 | 78.4 |
| USAD Large | 330M | 94.1 | 79.5 | 93.9 | 83.4 | 53.0 | 87.4 | **100.0** | 97.6 | 79.1 | 38.4 | 97.4 | 96.1 | 83.2 | 57.0 | 97.0 | 97.5 | 18.5 | 75.3 | 79.4 |
| $\text{SPEAR}_{s+a}$, Base | 95M | 92.4 | 78.7 | 93.5 | 83.8 | 49.9 | 86.5 | 96.9 | 95.2 | 71.9 | 24.6 | 96.8 | 94.1 | 78.9 | 64.6 | 96.9 | 97.0 | 21.9 | 77.3 | 77.8 |
| $\text{SPEAR}_{s+a}$, Large | 327M | 94.9 | 81.4 | 93.8 | 85.1 | 51.4 | 87.6 | 96.4 | 94.1 | 76.2 | 26.9 | 96.8 | 96.0 | 80.0 | 64.8 | 97.2 | 97.2 | 22.6 | 83.9 | 79.3 |
| $\text{SPEAR}_{s+a}$, XLarge | 600M | 95.3 | 81.6 | 95.4 | 85.1 | 52.4 | 88.6 | 98.5 | 96.3 | 78.7 | 28.0 | 97.0 | 96.3 | 81.5 | 65.1 | 97.2 | 98.1 | 22.6 | 83.6 | 80.1 |
| $\text{SPEAR}_{s+a}$, Base | 95M | 95.3 | 82.1 | 94.9 | 85.9 | 54.2 | 88.8 | **100.0** | 95.2 | 76.9 | 39.7 | 97.1 | 96.0 | 82.6 | 69.4 | 97.3 | 98.2 | 24.6 | 85.6 | 80.6 |
| $\text{SPEAR}_{s+a}$, Large | 327M | 94.9 | **83.8** | 95.9 | 87.4 | 56.4 | 89.2 | 99.2 | 97.6 | 78.9 | 40.0 | 97.4 | 97.5 | 85.3 | 70.2 | 98.1 | 98.3 | 25.7 | 88.5 | 81.8 |
| $\text{SPEAR}_{s+a}$, XLarge | 600M | 95.8 | 83.6 | 96.0 | 89.4 | 57.1 | **91.0** | **100.0** | 96.9 | 80.9 | 41.1 | 97.4 | 97.9 | 86.0 | 74.2 | **98.4** | **98.6** | **26.6** | **90.4** | **82.6** |

- **Teacher Model Selection**: We compared pre-training with MVQ tokens extracted from different SSL teacher models (see Appendix F.1.1) and different layers of teacher models (see Appendix F.1.2).

- **Masked Prediction Pre-training Loss**: We investigated how to balance the pre-training loss on the masked and unmasked positions for SPEAR (see Appendix F.2).

- **Encoder Architectures**: We compared using Zipformer and Transformer as the encoder backbone for SPEAR (see Appendix F.3).

- **MVQ Tokens vs k-means Tokens**: We compared using fine-grained MVQ tokens and k-means tokens as pre-training targets (see Appendix F.4). Additionally, we compared the feature subspaces reconstructed by the MVQ quantiser and a k-means clustering model (see Appendix F.4.2).

- **Number of Codebooks**: We studied the effect of varying the number of codebooks in the MVQ quantiser on the pre-training performance (see Appendix F.5).

## F.1. Teacher Models and Layers

### F.1.1. TEACHER MODEL SELECTION

In this group of ablation studies, different choices of SSL models are compared for generating pre-training targets under the SPEAR framework. In addition to WavLM Large and Dasheng 1.2B as presented in Section 4, HuBERT-Large (Hsu et al., 2021) and ATST-frame (Li et al., 2024), which are pre-trained with different SSL objectives and data scales, are used for generating fine-grained discrete targets for speech and audio data, respectively.

**HuBERT**  HuBERT (Hsu et al., 2021) is a speech SSL model pre-trained using masked language modelling (MLM) loss on 60k hours of speech from Libri-light (Kahn et al., 2020). In contrast to WavLM Large, HuBERT-Large is pre-trained on less diverse data (only read speech) without augmentations, resulting in weaker overall performance, especially on speaker-related tasks. A comprehensive comparison of HuBERT Large and WavLM Large on SUPERB can be found in Table 13 and Table 14. The representation from the 21st layer of HuBERT Large is used to train an MVQ quantiser with

*Table 17.* Performance on the SUPERB benchmark of speech-domain models trained with MVQ tokens extracted from different models. Best results in **bold**, 2nd best results are underlined. Token mixing is not used in all models.

| Model | # Params | Targets | Pre-train Data | SUPERB | | | | | | |
|---|---|---|---|---|---|---|---|---|---|---|
| | | | | ASR↓ | KS↑ | IC↑ | SV↓ | SID↑ | SD↓ | ER↑ |
| ***Speech SSL Models*** | | | | | | | | | | |
| HuBERT Large | 317M | - | 60k | 3.62 | 95.29 | 98.76 | 5.98 | 90.33 | 5.75 | 67.62 |
| WavLM Large | 317M | - | 94k | 3.44 | 97.86 | 99.31 | 3.77 | **95.49** | 3.24 | 70.62 |
| ***Ours*** | | | | | | | | | | |
| LARGE-H-1 | 327M | HuBERT Large | 50k | 3.24 | 97.05 | 99.47 | 4.11 | 91.52 | 3.84 | 69.86 |
| LARGE-H-2 | 327M | HuBERT Large | 84k | **3.17** | 97.79 | **99.51** | 3.49 | 94.42 | 3.24 | 70.91 |
| SPEAR$_S$ Large | 327M | WavLM Large | 84k | 3.27 | **97.89** | 99.47 | **3.14** | 95.55 | **3.2** | **72.1** |

16 codebooks for generating the pre-training targets. We train the following two Large models with pre-training targets generated from the HuBERT Large:

- LARGE-H-1: Pre-trained on Libriheavy **without** data augmentation. This model is used to contrast with HuBERT Large.

- LARGE-H-2: Pre-trained on Speech-84k using the WavLM-style data augmentation.

The pre-training performance is evaluated on SUPERB, and the results are shown in Table 17. Note that the SPEAR$_S$ Large model in Table 17 is also pre-trained with WavLM-style data augmentation. As can be seen, MVQ tokens extracted from HuBERT Large are also effective pre-training targets. LARGE-H-1 outperforms its teacher HuBERT Large on all SUPERB tasks by a large margin on the premise of using the same amount of pre-training data. Notably, LARGE-H-1 demonstrates strong ASR performance, achieving the lowest WER on SUPERB ASR task, even surpassing SPEAR$_S$ Large trained with more data, suggesting that the MVQ tokens extracted from HuBERT Large could have a stronger focus on semantic information (Mousavi et al., 2025).

By increasing the amount of pre-training data, LARGE-H-2 further improves over LARGE-H-1. However, LARGE-H-2 yields a weaker overall performance on SUPERB compared to SPEAR$_S$ Large, which is pre-trained with MVQ tokens extracted from WavLM Large. This suggests that MVQ tokens extracted from stronger speech representations translate to a stronger pre-training performance under SPEAR framework.

**ATST-frame** ATST-frame (Li et al., 2024) is an audio SSL model pre-trained with BYOL (Grill et al., 2020) objective on 5k hours of AS-2M with 86M parameters. The model generates 768-d frame-level audio representations at a 25Hz frame rate. We train the following two Base models for comparison:

- BASE-AUDIO-1: Pre-trained on AS-2M with MVQ tokens extracted from the last layer of ATST-frame using 8 codebooks.

- BASE-AUDIO-2: Pre-trained on AS-2M with MVQ tokens extracted from the last layer of Dasheng 1.2B using 8 codebooks.

The results of AT fine-tuning on AudioSet and HEAR benchmark are presented in Table 18.

As can be seen in Table 18, BASE-AUDIO-1 exhibits very strong performance on AudioSet AT tasks, achieving higher mAP than its teacher ATST-frame. By yielding an mAP of 40.3 on AS-20k and 49.6 on AS-2M, BASE-AUDIO-1 outperforms EAT (Chen et al., 2024a), setting a new state-of-the-art for audio SSL models pre-trained only on AS-2M. This validates the effectiveness of SPEAR as an audio SSL approach, as it shows that the student model can consistently outperform its teacher model used for generating the pre-training targets.

However, despite achieving a higher mAP on AudioSet, BASE-AUDIO-1 shows weaker generalisation capability than BASE-AUDIO-2, the model pre-trained with MVQ tokens from Dasheng-1.2B. This is shown by its lower performance on

*Table 18.* Performance of audio-domain models pre-trained with MVQ tokens extracted from different teacher models on AudioSet AT tasks and HEAR. All results are the higher the better. *: For fair comparison with ATST-frame, HEAR evaluation is performed using the concatenation of all layers' representations. Best results in **bold**. Token mixing is not used.

| Model | # Params | Targets | Pre-train Data | AudioSet | | HEAR* | | | | | |
|---|---|---|---|---|---|---|---|---|---|---|---|
| | | | | AS-20k | AS-2M | ESC | FSD | GZ-Gen | NS-5 | LC | SC-5 |
| *Audio SSL Models* | | | | | | | | | | | |
| EAT (Chen et al., 2024a) | 86M | - | 5k | 40.2 | 48.6 | - | - | - | - | - | - |
| ATST-frame | 88M | - | 5k | 39.0 | 48.0 | 89.0 | 55.7 | 88.3 | 68.6 | 78.1 | 92.6 |
| *Ours* | | | | | | | | | | | |
| BASE-AUDIO-1 | 94M | ATST-frame | 5k | **40.3** | **49.6** | **89.4** | **57.2** | 89.5 | 64.4 | 79.4 | **94.3** |
| BASE-AUDIO-2 | 94M | Dasheng 1.2B | 5k | 39.1 | 49.3 | 88.9 | 56.6 | **90.1** | **72.2** | **81.2** | **94.3** |

HEAR tasks from the speech and music domains (e.g., GZ-Gen, NS-5, LC, and SC-5). We attribute this to the fact that the Dasheng 1.2B model produces more generic audio representations due to the vast amount of pre-training data and enormous model size, and this quality is encapsulated in the MVQ tokens derived from the MVQ quantiser. This suggests that the choice of teacher model plays a critical role in our framework's pre-training quality, as high-quality, generic features can be transferred to the student via the MVQ tokens, even when the student is trained on significantly less data.

**Conclusion**   From Table 17 and Table 18, we conclude that the performance of SPEAR depends on the choices of teacher models for generating the pre-training targets. Under both speech-domain and audio-domain experiments, using a more powerful and generic teacher for pre-training targets generation leads to a better student model, indicating the necessity of using better teacher models for optimal performance. We also show that the performance of SPEAR framework is not upper-bounded by the teacher model, as our student models are always capable of outperforming their corresponding teacher model for generating the pre-training targets.

### F.1.2. TEACHER LAYER SELECTION

In this ablation study, experiments are carried out to compare different teacher layers for extracting pre-training targets. In our initial setup, the teacher layer is selected based on the downstream fine-tuning performance, i.e., ASR for speech teacher and AT for audio teacher. All experiments are conducted using the Base architecture, and the results are shown in Table 19.

For the speech teacher, we evaluate the 18th, 21st, and 24th (last) layers of WavLM Large and report the WERs on LS-100 fine-tuning tasks. The 21st layer is selected since it achieves the lowest WERs. Note that this choice aligns with the findings of Shi et al. (2024b), who also report the discrete tokens derived from the 21st layer of WavLM Large to yield strong ASR performance. For the audio teacher, we compare using the 25th, 35th, and 40th (last) layers of Dasheng-1.2B and report the mAP on the AS-20k fine-tuning task. The 40th layer is selected since it yields the highest mAP.

*Table 19.* Layer selection for speech teacher (WavLM) and audio teacher (Dasheng). Token Mixing is not used.

*(a)* WavLM layer comparison on LS-100 (WER).

| WavLM layer | test-clean ↓ | test-other ↓ |
|---|---|---|
| 18 | 3.1 | 6.1 |
| 21 (adopted) | **3.0** | **5.8** |
| 24 | 3.1 | 6.0 |

*(b)* Dasheng layer comparison on AS-20k (mAP).

| Dasheng layer | mAP ↑ |
|---|---|
| 25 | 38.4 |
| 35 | 38.7 |
| 40 (adopted) | **39.2** |

### F.2. Pre-training Loss

The hyperparameter $\alpha$ in Equation 1 controls the contribution of the prediction loss on the masked and unmasked frames. To investigate its influence w.r.t SPEAR, we conducted experiments on single-domain models, varying $\alpha$ from 0.0 (predicting only unmasked frames) to 1.0 (predicting only masked frames). The speech-domain models are pre-trained on LibriSpeech, while the audio models are pre-trained on AS-2M. The same MVQ tokens from Section 4 are used. We evaluate the downstream fine-tuning performance on LS-100 fine-tuning and AT, results presented in Figure 3. Note that token mixing is

not applied in this ablation.

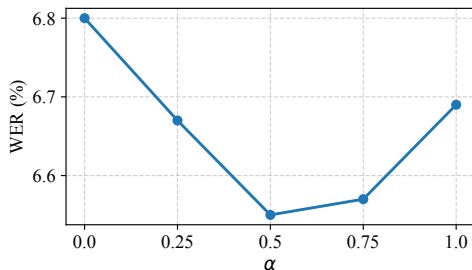 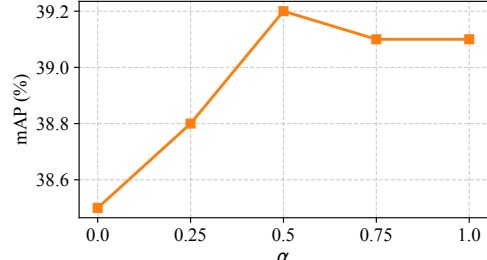

*Figure 3.* Effect of $\alpha$ on two downstream fine-tuning tasks. Left: WERs of test-other on LS-100 ASR fine-tuning task; Right: mAP on AudioSet evaluation set on AS-balanced fine-tuning task.

As shown in Figure 3, a balanced contribution of prediction loss on masked and unmasked frames with $\alpha = 0.5$ yields the best downstream performance. This observation diverges from the findings in HuBERT (Hsu et al., 2021), where computing prediction loss merely on masked frames (i.e. $\alpha = 1.0$) was optimal. We hypothesise this difference stems from the fine-grained nature of the MVQ tokens. Compared to the coarse units generated from k-means clustering, predicting fine-grained MVQ tokens is a significantly more challenging pretext task. Including the easier objective of predicting tokens at unmasked positions helps to regularize the model and stabilize the learning process. However, the pretext task must remain sufficiently challenging: setting $\alpha = 0.0$ makes the objective too simple, degrading it to a non-contextual prediction task that is ineffective for learning powerful representations. Thus, a balanced $\alpha$ is crucial for the success of the SPEAR framework.

### F.3. Encoder Architecture

To decouple the influence of the encoder architecture from our pre-training framework, this ablation compares the Zipformer (utilising filterbank inputs) against a standard Transformer backbone (utilising waveform inputs). The Transformer implementation follows the architecture adopted in WavLM (Chen et al., 2022). We pre-train both models on the full LibriSpeech 960h corpus (LS-960) for 300k updates, maintaining the same MVQ-based pre-training objective. The results on LS-100 ASR fine-tuning and the SUPERB benchmark are presented in Table 20.

*Table 20.* Comparison of Zipformer and Transformer as encoder backbone. Both models are trained without token mixing.

| Encoder Backbone | LibriSpeech finetune | | SUPERB | | | | |
|---|---|---|---|---|---|---|---|
| | test-clean | test-other | PR ↓ | IC ↑ | KWS ↑ | SID ↑ | ER ↑ |
| Transformer | 4.1 | 9.0 | **3.4** | 97.79 | **97.27** | **89.8** | 66.27 |
| Zipformer | **3.1** | **7.0** | **3.4** | **98.37** | 96.83 | 88.4 | **68.29** |

On the SUPERB tasks, which evaluate frozen representations, both architectures exhibit comparable performance: the Transformer proves stronger on SID, whilst the Zipformer excels on ER. However, the Zipformer demonstrates a distinct advantage in downstream ASR fine-tuning, a result consistent with its ASR-centric design (Yao et al., 2024). Moreover, the Zipformer is more computationally efficient due to its intermediate downsampling operations. For instance, pre-training the Zipformer requires approximately 350 GPU hours, roughly 60% of the 600 hours required for the Transformer. Consequently, the stronger ASR fine-tuning performance and computational efficiency motivate our selection of the Zipformer as the backbone architecture for SPEAR.

### F.4. MVQ Tokens and k-means Tokens

#### F.4.1. QUANTITATIVE EVALUATION

To isolate the impact of the quantization target, we conduct a controlled ablation comparing our MVQ tokens against standard k-means tokens. Both target types are derived from the same layer (the 21st layer) of WavLM Large, with the

k-means baseline utilising 2000 clusters. Using the Base architecture, both models are pre-trained for 300k updates on LibriSpeech under identical augmentation strategies (noise mixing, utterance mixing, and masking). Notably, for the k-means experiment, we adopt the standard configuration by computing the loss solely on masked positions (Hsu et al., 2021). The results on the LS-100 ASR fine-tuning task and the SUPERB benchmark are shown in Table 21.

*Table 21.* Comparison of MVQ tokens and k-means as pre-training target. Both models are trained without token mixing.

| Target | LibriSpeech finetune | | SUPERB | | | | |
|---|---|---|---|---|---|---|---|
| | test-clean | test-other | PR ↓ | IC ↑ | KWS ↑ | SID ↑ | ER ↑ |
| k-means, 2000 clusters | 3.5 | 7.2 | 4.0 | 97.92 | 96.79 | 86.6 | 67.56 |
| MVQ, 16 codebooks | **3.1** | **7.0** | **3.4** | **98.37** | **96.83** | **88.4** | **68.29** |

It can be seen that the model trained with MVQ tokens achieves better performance on all tasks, especially the two paralinguistic tasks: SID and ER. This aligns with our findings in Appendix F.4.2, where it is shown that the fine-grained MVQ tokens retain richer paralinguistic information than the coarse k-means tokens.

### F.4.2. FEATURE SUBSPACES ANALYSIS

In order to investigate if the codebooks in the MVQ quantiser have captured useful characteristics from the speech and audio representations, we visualise the reconstructed embedding space of the MVQ quantiser on a 2-D plane using UMAP (McInnes et al., 2018). Specifically, we visualise the speaker embeddings encoded by the MVQ quantiser of 10 speakers randomly drawn from LibriSpeech dev-clean sets, with each speaker having 25 utterances. The speech MVQ quantiser from Sec 4 is used. The speaker embeddings are computed with the procedure described below. First, we use the speech MVQ quantiser (see Table 4) to encode the frame-level embeddings generated by WavLM Large into the MVQ tokens. Then, we compute the reconstructed frame-level embeddings by summing over the encoded code vector from each codebook. The speaker embedding for each utterance is obtained by calculating the mean embedding vector over all frames. As a comparison, we also visualize the speaker embedding space represented through a k-means clustering model. The 500-cluster k-means model used for generating the pre-training targets for WavLM Large is adopted, which is trained on the 9th layer representations of HuBERT Base. We use the cluster centroid to represent each frame and average the frame-level embeddings along the temporal dimension to obtain a single speaker embedding for each utterance.

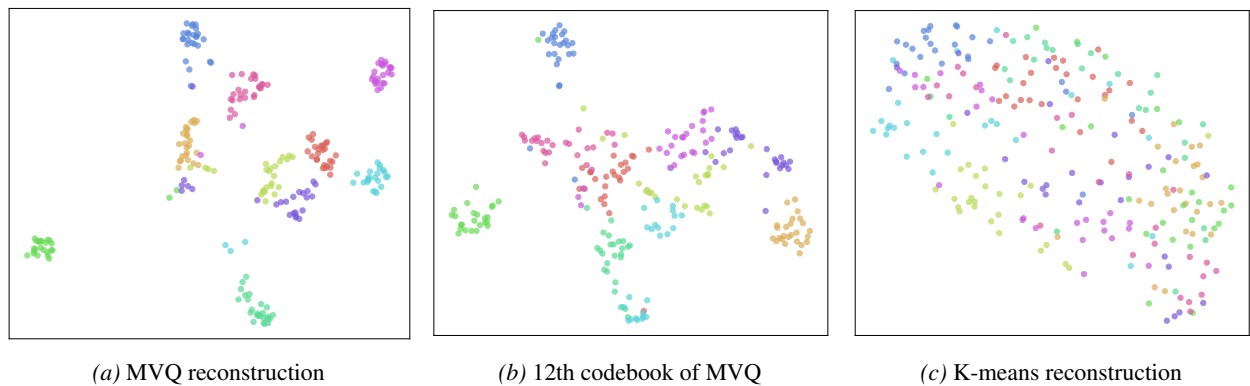

*(a)* MVQ reconstruction    *(b)* 12th codebook of MVQ    *(c)* K-means reconstruction

*Figure 4.* Comparing the reconstructed embedding space obtained through different MVQ quantisation and k-means. The speaker embeddings of 10 speakers drawn from LibriSpeech dev-clean are visualized using UMAP on a 2D plane, with each colour representing a single speaker. (a): Reconstruction using all codebooks of MVQ; (b): Reconstruction using only the 12th codebook of MVQ; (c): Reconstruction using k-means centroids.

The visualisations are shown in Figure 4. As can be seen, the MVQ quantiser successfully retains speaker characteristics, showing a clear separation between different speakers. It is noteworthy that a single codebook with 256 codes is capable of capturing certain levels of the speaker characteristics, showing reasonable separation between different speakers in Figure 4b. However, the k-means centroids fail to distinguish different speakers, showing poor separation for different speakers. This observation aligns with the fact that our speech-domain model SPEAR$_S$ Large achieves far better performance on SV compared to WavLM Large, a task requiring distinguishing different speakers by comparing the speaker embedding

similarity.

## F.5. Number of Codebooks

The relationship between the number of codebooks $N$ and the pre-training performance is investigated here. Experiments are carried out under single-domain settings with $N$ varying from 4 to 16, using the Base size model.

*Table 22.* Results of SPEAR speech-domain pre-training with different numbers of codebooks $N$. Best results in **bold**. All models are trained without token mixing.

| $N$ | LS-100 | | SUPERB | |
|---|---|---|---|---|
| | test-clean↓ | test-other↓ | SID↑ | ER↑ |
| 4 | 3.34 | 7.01 | 83.12 | 67.24 |
| 8 | 3.19 | 6.77 | 84.83 | 67.76 |
| 16 | **3.08** | **6.55** | **86.35** | **68.29** |

For speech-domain experiments, the same representations from the 21st layer of WavLM Large are used to train the MVQ quantiser. We pre-train the models on LS-960 for 300k updates. The performances on the following three tasks are evaluated: ASR fine-tuning on LS-100, speaker identification (SID), and emotion recognition (ER) from SUPERB, which serve as indicators of the model's understanding and paralinguistic capabilities. The results are shown in Table 22. As can be seen, increasing the number of codebooks for speech pre-training consistently enhances the model performance. The WER on the test-other set is reduced by 6.4% with $N$ increasing from 4 to 16. Moreover, models trained with a larger $N$ also exhibit stronger paralinguistic capabilities, which are manifested through their performance on SID and ER. This implies that increasing $N$ to 32 could lead to further performance improvement for speech-domain models.

*Table 23.* Results of audio-domain pre-training with different numbers of codebooks $N$. All results are the higher the better. Best results in **bold**. All models are trained without token mixing.

| $N$ | AudioSet | | HEAR | | | |
|---|---|---|---|---|---|---|
| | AS-20k | AS-2M | Environment | Speech | Music | Average |
| 4 | **39.2** | 49.1 | 77.63 | 68.09 | 80.30 | 75.63 |
| 8 | **39.2** | **49.3** | **80.33** | 69.87 | **80.70** | **77.01** |
| 16 | 38.9 | 49.0 | 80.25 | **69.92** | 80.64 | 76.97 |

Similar experiments are conducted for audio-domain pre-training. Following Section 4, the last layer of Dasheng 1.2B is used to train the MVQ quantiser with 4, 8, and 16 codebooks. The models are evaluated on the AudioSet fine-tuning task, and the results are shown in Table 23. As can be seen, increasing $N$ from 4 to 8 improves the downstream AT fine-tuning performance and the HEAR scores. However, further increasing $N$ to 16 degrades the pre-training performance, leading to a lower mAP and average HEAR score compared to $N = 8$. We hypothesise that representations of audio SSL models encapsulate less information compared to speech representations in general. Therefore, using a moderate number of codebooks seems to be enough for audio-domain pre-training. A too large $N$ might force some codebooks to capture the nuances in the audio teacher representations and introduce noise to the pre-training.

## F.6. Loss Weighting

*Table 24.* Effect of $\lambda$ in dual-domain pre-training. All models are trained without token mixing.

| $\lambda$ | LS-100 | | AS-20k |
|---|---|---|---|
| | test-clean ↓ | test-other ↓ | mAP ↑ |
| 0.3 | 3.0 | 5.9 | **37.0** |
| 0.2 | 3.0 | 5.7 | 36.9 |
| 0.1 | **2.9** | **5.6** | 36.9 |

As shown in Section 3.2.2, the hyperparameter $\lambda$ controls the contribution of the general-audio masked-prediction loss during joint training. To determine the optimal balance, we conduct an ablation study with 3 values of $\lambda$ using our SPEAR Large architecture and compare the fine-tuning performance on LS-100 and AS-20k after 100k pre-training steps. The experimental results are shown in Table 24. We observed that reducing $\lambda$ from 0.3 to 0.1 yields a 0.3 absolute WER improvement on test-other, while the mAP is only reduced by 0.1 absolute. Consequently, we adopted $\lambda=0.1$ in our dual-domain experiments for a balanced performance across both domains.

## G. Comparison with USAD

In this section, we performed a controlled comparison between SPEAR and USAD (Chang et al., 2025), another framework for joint speech and audio representation learning, also leveraging multiple domain-specific teachers. Specifically, we trained a new dual-domain SPEAR model with the Base architecture, named SPEAR (USAD-aligned), mirroring the USAD settings. We used the same teacher models as used in USAD, namely WavLM Base+ (speech) and ATST-Frame (Li et al., 2024) (audio) to extract the MVQ tokens as pre-training targets in SPEAR, aiming to rule out the effect of stronger teacher models being used in SPEAR. We also used a subset of the USAD training corpora, excluding Fisher and VoxLingua for speech and SoundNet for audio, due to availability issues. A summary of the model configurations and data usage for both models is shown in Table 25.

Table 25. Model configurations of SPEAR (USAD-aligned) and USAD. Approximate data amount in hours.

| Model | # Params | Speech data | Audio data | Total data |
|---|---|---|---|---|
| SPEAR (USAD-aligned) | 94M | 86k | 9.3k | 95.3k |
| USAD Base | 94M | 91k | 35k | 126k |

Table 26. Comparison between SPEAR (USAD-aligned) and USAD on SUPERB and HEAR. HEAR results for both SPEAR (USAD-aligned) and USAD Base are obtained with feature concatenation. Token mixing was not applied for SPEAR for fair comparison.

| Model | # Params | Data | SUPERB | | | | | | HEAR | | | |
|---|---|---|---|---|---|---|---|---|---|---|---|---|
| | | | PR ↓ | ASR ↓ | IC ↑ | KS ↑ | SID ↑ | ER ↑ | Env ↑ | Speech ↑ | Music ↑ | Avg ↑ |
| SPEAR (USAD-aligned) | 94M | 95.3k | **4.6** | **5.1** | **98.7** | **97.4** | **89.2** | **69.4** | **81.1** | **76.9** | **80.5** | **79.4** |
| USAD Base | 94M | 126k | 5.1 | 7.7 | 98.3 | 97.1 | 88.6 | 68.0 | 80.7 | 73.7 | 79.3 | 77.8 |

The comparison between SPEAR (USAD aligned) and USAD on SUPERB and HEAR is shown in Table 26. As can be seen, SPEAR (USAD-aligned) consistently outperforms USAD Base, despite only using a subset of the training data used by USAD, suggesting that SPEAR is more effective than USAD for learning unified speech and audio representations. We attribute this performance gap to the following two reasons:

- **Training objectives**: In SPEAR, the student is trained to predict the discrete tokens extracted from teacher models given a masked input, which is a frequently used pretext task for SSL. This combination of KD and SSL in SPEAR enables the student to learn generic representations while benefiting from the knowledge of the two domain-specific teachers, creating a student model even with the capability of surpassing teacher models (e.g. SPEAR$_S$ Large outperforms WavLM Large). On the other hand, USAD enforces the student to mimic the teacher representations through L1 and cosine distance loss. Consequently, the student performance is theoretically upper-bounded by the teacher's performance.

- **Joint feature matching is ill-defined for disparate domains**: In USAD, the student effectively minimises the distance to two embedding spaces (speech and audio) simultaneously. The L1 losses induced by two teachers encourage the student model to find a "mean" of two representation spaces. Since the feature spaces could be distinct, this "mean" representation may lie in a region of the manifold that lacks semantic meaning for either domain. SPEAR avoids this risk by quantising representations into discrete tokens via MVQ, where the tokens exhibit the capability of representing a certain characteristic of the input speech/audio data (see Appendix F.4.2, where we found a single codebook to contain rich speaker information). This allows the model to retain distinct, high-fidelity details by learning to predict the discrete tokens for both domains simultaneously, with lower risk of destructive interference.

