# OpenReview forum: "SPEAR: A Unified SSL Framework for Learning Speech and Audio Representations"
_ICML.cc/2026/Conference — ICML 2026 regular_

### Official Review · Reviewer_H3i4 · 2026-03-03

**Soundness:** 3
**Presentation:** 2
**Significance:** 3
**Originality:** 3
**Overall Recommendation:** 4
**Confidence:** 3

**Summary:**

The paper presents SPEAR, a unified self-supervised learning framework designed to bridge the gap between speech and general audio representation learning. The core innovation lies in using Multi-codebook Vector Quantization to convert continuous teacher representations into fine-grained discrete tokens, which then serve as targets for masked token prediction. This approach allows a single student model to inherit complementary knowledge from domain-specific specialists.

**Compliance With Llm Reviewing Policy:**

Affirmed.

**Final Justification:**

Thank you for your response and explanation. I am willing to raise my score.

**Key Questions For Authors:**

1. While the student model can surpass its teachers, its performance remains heavily dependent on the quality of the selected domain-specific teachers.
2. Is it feasible to train a unified model from scratch using all combined speech and audio data following the Best-RQ or HuBERT methodologies? If the exact same combination of speech and audio data is used to train a WavLM-style model, how would it perform compared to SPEAR?
3. Why is the concatenation of all layers' features only utilized in Table 5? Is this same strategy applied in Table 3 and Table 4? Does the reliance on concatenated features in Table 5 imply that they yield superior performance compared to single-layer or weighted-sum representations?

**Strengths And Weaknesses:**

1. The introduction of MVQ for SSL pre-training is novel. By producing exponentially more representable states than coarse methods like k-means, it preserves the acoustic and temporal details necessary for both speech and audio tasks.
2. The proposed asymmetrical dual-domain objective intelligently handles the inherent differences between domains by using speech tokens as universal targets while limiting audio targets to general audio data.
3. The token mixing mechanism stochastically combines targets from multiple sources, improving performance in complex, overlapping sound scenes.

---

> ### Author Rebuttal · Authors · 2026-03-31
>
> We thank the reviewer for acknowledging the novelty of SPEAR, including the introduction of MVQ for SSL pre-training, the asymmetrical dual-domain strategy and the token mixing mechanism. Below, we provide response to the questions raised by the reviewer.
>
> ### Q1: Dependency on teacher quality
>
> We agree that SPEAR depends on the quality of the domain-specific teachers: stronger teachers generally produce higher-quality MVQ targets, which in turn lead to stronger students. This is a genuine trade-off of the framework. However, SPEAR is not constrained to merely reproduce the teacher performance. As shown in Tables 17 and 18, SPEAR trained with HuBERT Large or ATST-frame targets can still outperform the corresponding teacher models, demonstrating that the **MVQ-based masked prediction objective allows the student to learn beyond direct imitation**. This is an important distinction from feature-matching KD approaches, where the student is much more tightly tied to the teacher representation space.
>
> ### Q2: Can a unified model be trained from scratch on mixed speech+audio data?
>
> To address this question, we trained a HuBERT-style two-iteration baseline on the same mixed speech+audio corpus (SA-97k) used by SPEAR_{s+a} Base. In the first iteration, we applied k-means (256 clusters) to MFCC features; in the second, we trained another k-means (2000 clusters) on the 6th-layer representations of the first-iteration model. The encoder backbone, training data, and batch size were matched to SPEAR_{s+a} Base for a fair comparison.
>
> | Model | # Params |PR↓|ASR↓|SID↑|ER↑|Env↑|Speech↑|Music↑|Avg↑|
> | --- | --- | --- | --- | --- | --- | --- | --- | --- | --- |
> | HuBERT-style | 94M | 4.0 | 4.8 | 88.9 | 67.7 | 61.4 | 69.5 | 63.5 | 64.8 |
> | SPEAR_{s+a} Base | 94M | **3.8** | **3.8** | **90.3** | **69.2** |**80.6** | **74.0** | **73.9** | **77.8** |
>
> This controlled experiment shows that HuBERT-style SSL trained from scratch on mixed speech+audio data performs substantially worse than SPEAR, especially on the audio-side HEAR metrics. **This supports our central claim that unifying speech and audio is non-trivial, and that distilling from domain-specialised teachers is highly beneficial.**
>
> More broadly, this is consistent with prior observations on other SSL objectives. We refer the reviewer to the data2vec[1] experiment reported in USAD[2] (their Table 3), where the data2vec-style SSL training on combination of speech and audio data also fails to produce strong unified representations. Together, these results suggest that simply scaling a single SSL objective to mixed speech+audio data is insufficient, whereas SPEAR’s teacher-guided formulation is much more effective.
>
> ### Q3: Why is layer concatenation used only in Table 5?
>
> In Table 5 (HEAR), we report results using both the last-layer representation and the concatenation of all layers. The concatenated setting is included only for fair comparison with USAD, since USAD only reports HEAR results obtained with layer concatenation (personal communication from USAD first author). We also report the last-layer results of SPEAR in the same table to ensure fair comparison with the other baselines.
>
> As for other experiments, Table 4 (SUPERB) follows the standard SUPERB protocol, which uses a weighted sum of intermediate layers; all models in Table 4 use that same protocol, as stated in Section 5.2. Table 3 reports full fine-tuning on ASR and AT, where the encoder itself is updated together with the downstream head. Therefore, the use of concatenated features in Table 5 does not imply that they are universally superior; it is included there solely to align with the evaluation protocol used by USAD, the only dual-domain baseline.
>
> We would also like to provide the results of an additional SPEAR_{s+a} XLarge model trained with an extra 13k hours of audio-domain data compared to the original SPEAR_{s+a} in the paper. Without layer-concatenation, this model achieves a higher average score on HEAR than the same-sized Dasheng-0.6B while being more versatile, since Dasheng achieves poor results on ASR or speech-focused SUPERB.
>
> | Model | # Params | Data (hours) | Env | Speech | Music | Avg |
> | --- | --- | --- | --- | --- | --- | --- |
> | Dasheng-0.6B | 600M | 272k | **83.0** | 74.8 | **84.7** | 81.0 |
> | SPEAR_{s+a} XLarge | 600M | 197k | 82.3 | **77.0** | 81.3 | 80.1 |
> | SPEAR_{s+a} XLarge, more audio data | 600M | 210k | 81.9 | **77.0** | 84.1 | **81.1** |
>
> [1]: Baevski, Alexei, et al. "Data2vec: A general framework for self-supervised learning in speech, vision and language." Proc. ICML 2022
>
> [2]: Chang, Heng-Jui, et al. "USAD: Universal speech and audio representation via distillation." Proc. ASRU 2025.

---

> > ### Author Rebuttal · Reviewer_H3i4 · 2026-04-03
> >
> > Thank you for the detailed response. My main worry is the 'ceiling effect' inherent in distillation. When the teacher model's efficacy is insufficient, the distilled model's capacity for improvement is capped, potentially hindering the exploration of performance levels beyond what the teacher can provide.

---

> > > ### Author Response · Authors · 2026-04-04
> > >
> > > Thank you for the constructive follow-up. We agree with the reviewer that teacher quality is an important factor in SPEAR, as in any knowledge distillation (KD) based methods: stronger teachers generally produce better supervision targets, which in turn lead to stronger students. We have already noted this dependence in the manuscript (Appendix F.1.1), and **we will make this limitation clearer in the revised version**.
> > >
> > > However, SPEAR is not limited to merely reproducing the teacher's performance. Unlike direct feature-matching KD, SPEAR combines KD with masked prediction of fine-grained MVQ tokens, so the student learns through a contextual SSL objective rather than direct imitation. In practice, **this allows SPEAR to surpass its teachers, as shown in our experiments with different teacher choices** (Appendix F.1.1). This is also consistent with our controlled comparison against USAD, another general-purpose encoder built with direct feature-matching KD methods, where SPEAR consistently performs better.
> > >
> > > More importantly, we believe distillation is particularly well motivated in this setting, because **unifying speech and general audio is non-trivial**. As shown in our rebuttal, a HuBERT-style model trained from scratch on the same mixed speech+audio data performs substantially worse than SPEAR, suggesting that **a single unified SSL objective is insufficient to bridge the domain mismatch between speech and general audio**. We therefore view teacher dependence as a deliberate trade-off that enables strong unified learning across these heterogeneous domains. In this sense, SPEAR provides a practical mechanism for combining strong speech-specialised and audio-specialised models into a single unified encoder, and can naturally benefit as stronger domain-specific teachers become available.
> > >
> > > ---
> > >
> > > **Update:**
> > >
> > > Thank you again for the helpful feedback. We hope our response clarifies why, although teacher dependence is a genuine trade-off, SPEAR is not merely limited to reproducing teacher performance, and why distillation is a well-motivated design choice for bridging the speech–audio domain mismatch. We hope this clarification is useful for your final evaluation of the paper.

---

### Official Review · Reviewer_cjHg · 2026-03-04

**Soundness:** 3
**Presentation:** 3
**Significance:** 3
**Originality:** 2
**Overall Recommendation:** 4
**Confidence:** 5

**Summary:**

The paper proposes to distill knowledge from two domain-specific audio SSL pre-trained encoders into a single encoder. This is achieved by incorporating the idea of multi-codebook vector quantization for generating training targets from the domain-specific models. Additionally, data augmentation via waveform and token mixing and training with asymmetric losses is performed. The distilled encoder successfully outperforms the original domain-specific encoders and shows good performance compared to other baselines.

**Compliance With Llm Reviewing Policy:**

Affirmed.

**Final Justification:**

The framing in the final reply to rebuttal comment by the authors, makes a lot of sense to me, with proper emphasis and key distinctions. In the final version, if this is included in the main claims, I would be happy to raise my score. While the same information can be decoded from reading the paper, including this framing explicitly makes the paper much more intuitive and readable, and better presents its strength.

**Key Questions For Authors:**

- The N independent linear prediction heads: do they all take the full $h_t$, or is it split into the $N$ subspaces as done for the teacher embedding MVQ?

- Why is a random delay used to mix $w$ and $w'$, since both are different random samples, isn’t directly mixing them enough?

- "Therefore, we apply both in-batch utterance mixing and noise mixing during training": is this in addition to the token mixing augmentation, or are the noise samples considered as the "audio" domain?

- "perform token mixing using the MVQ tokens generated by the speech SSL teacher": even for the audio domain? Are the mixing tokens extracted using only the speech encoder?

- "We hypothesise that representations of audio SSL models encapsulate less information compared to speech representations in general": I would have thought the opposite, since the audio domain is significantly more varied. I would have expected increasing the number of quantizers to potentially lead to better performance.

**Limitations:**

Limitations of the work are not discussed.

**Strengths And Weaknesses:**

### Strengths

- Logical combination and extension of existing techniques to achieve good results on ASR and AT tasks as well as the SUPERB benchmarks.

- Token mixing augmentation strategy is interesting.

### Weaknesses

- There seems to be limited new insights that could be taken from this work. The fact that multiple single-domain teachers could be used to train a single student for joint representation has already been shown in USAD (which is the main baseline in this paper). The use of MVQ for knowledge distillation has also already been explored in the original MVQ paper. The statement that the application of MVQ for masked SSL prediction is novel seems to be an overstatement. While the exact formulation may not exist, the use of MVQ for KD in the context of speech had already been explored in the original MVQ paper by Guo et al., 2023.
**Actionable:** The paper would benefit from a clearer articulation of what is fundamentally new compared to USAD and prior MVQ-based distillation approaches.

- This paper criticises USAD regarding the joint losses. However, even from the ablation study, the difference between the JOINT and ASSYMETRICAL is not significant. ASSYMETRICAL is only marginally better, which might even be due to training randomness. Different seeds might produce results that do not show statistical significance between these two paradigms.
**Actionable:** Please report results across multiple random seeds or provide variance to demonstrate whether the difference is statistically meaningful.

- The token mixing augmentation strategy, while interesting, seems somewhat arbitrary. While mixing in the audio domain makes sense, the token mixing assumes that signal power meaningfully dictates the distribution of the MVQ tokens for mixed signals, which is not clearly justified. Since the mixed waveform is already available, it is unclear why extracting MVQ tokens directly from the mixed signal would not be sufficient. Additionally, if token mixing is useful, it could potentially be applied more generally (e.g., mixing tokens from both teachers for all signals rather than only for mixed signals).
**Actionable:** Please provide stronger motivation or ablation studies comparing token mixing against simpler alternatives such as extracting MVQ targets directly from the mixed waveform.


- The comparison between USAD and SPEAR is probably not fair, since USAD was trained with 800 sec batches for 400k updates, while SPEAR is trained with 6.4k sec batches for 4–500k updates, which means SPEAR has seen more than 8 times the amount of data that USAD has seen. Maybe training USAD longer would perform at the same level. Similarly, for fair comparison even with the WavLM teacher models, the teacher model's performance might also improve if trained for another 2–300k iterations on this current dataset.
 **Actionable:** Please clarify the effective training budget (e.g., total audio seen or compute) and consider reporting results under more comparable training conditions.


Overall, the method appears well engineered and achieves strong empirical results. However, the conceptual and methodological insight beyond the combination of existing techniques remain somewhat limited.

---

> ### Author Rebuttal · Authors · 2026-03-31
>
> We thank the reviewer for recognising the strong performance of SPEAR on ASR, AT and SUPERB. Below we address the weaknesses and questions raised by the reviewer.
>
> ### W1: Differences to prior work
> We respectfully disagree that SPEAR is merely a combination of USAD and prior MVQ-based distillation.
>
> **USAD comparison:**
>
> **Feature-matching vs masked token prediction.** A fundamental difference is the optimisation objective. USAD uses feature matching in continuous space by training the student to minimise L1/cosine distance to teacher representations. With this objective, the best solution is to "fully imitate" the teachers i.e. the student is constrained by the teacher. SPEAR instead uses discrete MVQ targets in a masked-token prediction objective hence changing the problem from direct imitation to contextual SSL. **This allows SPEAR to learn stronger contextualised representations and even surpass its teachers (Appendix F.1.1)**. Also, USAD must align the student to two heterogeneous continuous embedding spaces simultaneously while SPEAR predicts discrete MVQ tokens of two teachers, reducing the risk of destructive interference.
>
> **Critical dual-domain designs.** SPEAR also introduces a novel asymmetrical dual-domain objective to handle domain mismatch and token mixing to preserve complete audio information in complex sound scenes, both are absent from USAD.
>
> **Original MVQ comparison:**
>
> A key distinction is SPEAR **predicts MVQ tokens given masked input for SSL**. MVQ was originally used for non-masked KD closer to regressing compressed teacher embeddings and it performed worse than continuous KD (L1/L2 loss). In SPEAR the student must predict MVQ targets from masked input, making this a contextual SSL task rather than direct KD, encouraging student to learn more powerful representations. Moreover, original MVQ is only explored for ASR while SPEAR focuses on unified SSL over speech and general audio, evaluating on 30+ downstream tasks.
> ### W2: Justification of Assymetrical strategy
> We respectfully disagree that the gain from the asymmetrical strategy is marginal. The 0.3 absolute improvement on HEAR (Table 7) is averaged over 18 tasks, so it reflects a broad improvement rather than a fluctuation. More importantly, we observed the same trend for both Base and Large models (not shown due to space), supporting the robustness of the asymmetrical design. This is also consistent with our motivation: speech tokens provide a more universal supervision while audio-specific targets are mainly beneficial for audio inputs.
> ### W3: Token mixing
> We did test extracting MVQ targets directly from the mixed input and it performed worse than token mixing because teacher models do not provide reliable targets for mixed signals. WavLM is trained with a denoising objective and tends to suppress the secondary source so that tokens extracted from the mixed signal fail to reflect both sources. Token mixing instead combines the clean-source targets within the same token space according to relative signal power, providing a simple but effective approximation that preserves supervision from both sources. This is particularly important for tasks such as sound source separation and, more broadly, for understanding complex acoustic scenes when extending to audio LLMs.
>
> We also note that mixing tokens across the speech and audio teachers is not well-defined: their MVQ quantisers are trained separately with non-shared codebooks, so their token indices belong to different discrete spaces.
> ### W4: Comparison with USAD
> We clarify that the SPEAR (USAD-aligned) model in Table 26 is trained for 200k updates with batch duration of 4.8k seconds. To match the total amount of seen audio with USAD (800-second batch, 400k updates), we report an earlier 50k-update SPEAR checkpoint. Under this controlled setting, SPEAR still outperforms USAD Base by a clear margin.
>
> |Model|# Params|Data (h)|Updates|PR↓|ASR↓|KS↑|SID↑|ER↑|Env↑|Speech↑|Music↑|Avg↑|
> |-|-|-|-|-|-|-|-|-|-|-|-|-|
> |SPEAR (USAD-aligned)|94M|95.3k|50k|**4.8**|**5.4**|**97.3**|**89.0**|**69.1**|**81.7**|**76.7**|**80.2**|**78.9**|
> |USAD Base|94M|126k|400k|5.1|7.7|97.1|88.6|68.0|80.7|73.7|79.3|77.8|
> ### Other questions
> Q1: Each prediction head takes the full $h_t$.
>
> Q2: The random delay simulates more diverse overlap patterns, e.g. partially mixed inputs, helping the model handle more realistic sound scenes.
>
> Q3: This is a typo, the "in-batch utterance mixing" here should be token mixing. In parallel, we apply noise mixing with high SNRs (10–20 dB).
>
> Q4: Yes, the mixing tokens are generated by the speech teacher. Under the asymmetrical strategy, the audio teacher does not induce a loss on speech data.
>
> Q5: We consistently found N=8 better for audio domain across two teachers. We hypothesise that audio representations are less structured than speech (e.g. unlike phoneme-based speech representations), so further increasing N makes the masked prediction task unnecessarily difficult.

---

> > ### Author Rebuttal · Reviewer_cjHg · 2026-04-03
> >
> > Thanks for the detailed response.
> >
> > My main conceern is that masked token prediction itself is not new in audio SSL. Methods such as wav2vec 2.0, HuBERT, and WavLM already use masked prediction with discrete targets, and HuBERT already follows an iterative refinement approach by performing k-means quantization on previous iteration of the model (teacher). So this general paradigm is already well established.
> >
> > The idea of using multi domain teachers is already present in USAD. So from my understanding, the main difference here is essentially replacing the direct KD of USAD with MVQ-based masked prediction targets.
> >
> > Also, the fact that MVQ targets are effective for KD has also already been shown in the original MVQ paper, even if the training setup is different, but it still shows the effectiveness of the MVQ methodology as strong teacher targets for KD.
> >
> > The empirical results are strong, but the core novelty seems more incremental in terms of combining and adapting existing ideas rather than introducing a fundamentally new SSL approach.

---

> > > ### Author Response · Authors · 2026-04-04
> > >
> > > We thank the reviewer for the thoughtful feedback and for recognising the strong performance of SPEAR. While individual components (masked prediction, multi-domain teachers) have appeared in isolation, SPEAR's novelty lies in the introduction of a paradigm shift from single-source representation to **multi-source, fine-grained, interference-aware learning under complex sound scenes in joint speech-and-audio SSL**, which is not addressed in prior work.
> > >
> > > **1. From single-source to explicit multi-source learning**
> > >
> > > Real-world environments often contain complex sound scenes with multiple overlapping speech, audio, and music sources. However, existing SSL methods are primarily designed for single-source data or suppress overlapped secondary sources (e.g. WavLM). SPEAR instead **proposes a stochastic token mixing mechanism to explicitly model complex sound scenes**, allowing the representation to preserve information from multiple overlapping sources. This is reflected in the strong performance on tasks involving complex sound scenes (e.g., speech separation, speaker diarisation, speech enhancement) in Table 6 and 14. To our knowledge, this is the first SSL framework that directly targets multi-source representation fidelity within the masked-token prediction paradigm.
> > >
> > > **2. From single-level targets to fine-grained multi-level supervision**
> > >
> > > Prior work, such as HuBERT, uses single-level k-means targets for masked prediction. By contrast, SPEAR uses MVQ tokens as **fine-grained multi-level supervision**. The independence between MVQ codebooks allows different codebooks to capture distinct speech/audio characteristics (e.g., linguistic, paralinguistic, environmental), rather than collapsing them into a single clustering space. Masked prediction in SPEAR is hence performed over a factorised target space, which differs fundamentally from prior iterative clustering in both granularity and structure. The strong performance of SPEAR on acoustics, linguistic and paralinguistic tasks (Appendix F.4) further validates the effectiveness of our novel integration of MVQ for speech-and-audio SSL.
> > >
> > > **3. Interference-aware domain fusion**
> > >
> > > While USAD introduces multi-domain teachers, it relies on direct feature distillation, which can be affected by domain interference (see Appendix G). SPEAR instead formulates domain fusion as a masked prediction problem over domain-specific MVQ tokens and proposes an asymmetrical training strategy to mitigate cross-domain interference. This leads to robust unified speech and audio modelling, further supported by our controlled comparison against USAD presented in the rebuttal.
> > >
> > > ### Summary
> > > In conclusion, we believe developing a general-purpose encoder capable of unifying speech and audio remains a critical yet underexplored problem. While MT2KD[1] initiated this direction by distilling from multimodal domain-specific supervised teachers, and USAD subsequently extended this to SSL-based teachers, SPEAR makes a further substantive advance by:
> > > - explicitly handling complex multi-source sound scenes,
> > > - factorising supervisions via fine-grained MVQ tokens,
> > > - reducing domain interference through masked-token prediction and asymmetrical loss.
> > >
> > > We therefore believe SPEAR represents a non-trivial advance in both problem formulation and training mechanism, rather than an incremental combination of existing techniques.
> > >
> > > [1]: Yang, Xiaoyu, et al. "MT2KD: Towards a general-purpose encoder for speech, speaker, and audio events." IEEE Transactions on Audio, Speech and Language Processing (2025).
> > >
> > > ---
> > > **Update:**
> > >
> > > Thank you again for the thoughtful feedback. We hope our clarification makes it clearer that the novelty of SPEAR lies not in the incremental combination of existing techniques, but in the non-trivial reformulation of unified speech-and-audio SSL through **explicit multi-source learning, fine-grained MVQ token prediction, and interference-aware domain fusion**. We hope this is helpful for your final evaluation of the paper.

---

### Official Review · Reviewer_qmYp · 2026-03-13

**Soundness:** 3
**Presentation:** 4
**Significance:** 3
**Originality:** 3
**Overall Recommendation:** 5
**Confidence:** 4

**Summary:**

The authors propose SPEAR, a framework for training universal audio encoders for speech, sound, and music. SPEAR combines distillation from domain-specific encoders (speech and non-speech), vector quantization, and masked language modeling. The final result is a single model that excels in encoding information across all audio domains, as shown in benchmark results on SUPERB and HEAR.

**Compliance With Llm Reviewing Policy:**

Affirmed.

**Final Justification:**

The authors clarified my initial concerns and mis-understandings in the rebuttal. This included the clear weaknesses and limitations of the approach, but I believe the empirical results and thorough ablations make this paper a valuable asset to the scientific community. The rebuttal therefore reinforced my prior assessment.

**Key Questions For Authors:**

1. How computationally/resource heavy is MVQ training?
2. I expect one weakness of SPEAR is increased computational cost from needing to obtain targets from multiple teachers. Approximately how long would an equivalent WavLM or HuBERT style model take to train on your setup (assuming same number of  training steps, batch size, model size etc) and ?

**Limitations:**

Yes

**Strengths And Weaknesses:**

Strengths:
- The authors present SPEAR, a novel method for creating a universal audio encoder for speech and non-speech. SPEAR combines distillation from domain-specific encoders (speech and non-speech), vector quantization, and masked language modeling.
- The authors conduct thorough experiments on the different design choices of their proposed architecture, showing that each proposed component (MVQ, distillation, encoder architecture) is well-justified. They also conduct many trials for selecting the optimal hyper-parameters.

Weakness:
- The main weakness of the method is the reliance on pre-existing models to perform distillation, including the need to train the MVQ offline. This introduces many hyperparameters that the authors must tune, such as the layer to obtain features from, the codebook size, etc.
- It is also reasonable to expect that results with the SPEAR framework will almost always yield stronger results than the models it builds upon, since it can be seen as another form of HuBERT's iterative approach. While the authors showed decent performance with other weaker SSL models in appendix ablations, I am curious if the strength of this framework will hold with an even weaker teacher (like MFCCs or even older SSL models).
- I think it would make more sense if the token-mixing ablation (Table 6) was done on audio benchmarks instead of speech. It currently shows little change in performance when compared to WavLM-style denoising, which should be expected since the downside of WavLM is only on non-speech audio. Strong results on non-speech audio will make it more clear if the proposed method is truly superior.

---

> ### Author Rebuttal · Authors · 2026-03-31
>
> We thank the reviewer for recognising the novelty of SPEAR in building a universal encoder for speech and audio, and the thouroughness of our experiments. Below, we address the weaknesses and questions raised by the reviewer.
>
> ### W1: Reliance on existing models
> We agree that SPEAR relies on existing strong SSL teachers and offline MVQ training, and this introduces additional design choices such as teacher layer selection and codebook configuration. This is indeed a trade-off of the framework. Our goal, however, is not to remove teacher dependence, but to show that knowledge from domain-specific speech and audio teachers can be distilled into a single unified encoder more effectively than prior approaches such as feature matching. We also note that the main design choices raised by the reviewer (e.g. teacher layer selection, number of codebooks, loss weighting) are explicitly studied in the appendix, where we show that they are well motivated rather than arbitrary.
>
> ### W2: Iterative improvement
> We agree that experiments with even weaker teachers would be interesting, but we see this as an extension rather than a requirement for the current paper. The main challenge addressed by SPEAR is not simply iterative refinement of a single teacher, but the unification of distinct speech and audio teachers within one student model, which is substantially different to HuBERT-style iterative training. Given that strong domain-specific SSL models already exist, we believe it is both practical and important to show that they can be effectively combined into a more generic and versatile encoder. We also note that Appendix F.1.1 already shows that SPEAR remains effective with weaker teachers such as HuBERT Large and ATST-frame.
>
> ### W3: Token mixing ablation
> Token mixing is designed to improve robustness in complex sound scenes, such as **overlapped speech** (speech mixed with speech) and **noisy speech** (speech mixed with audio). We therefore chose to evaluate it on three SUPERB tasks that directly reflect this setting: speaker diarisation (SD), speech separation (SS), and speech enhancement (SE). We believe these tasks are well suited to demonstrate the intended effect of token mixing. As shown in Table 6, token mixing outperforms WavLM-style denoising by a clear margin on SD and SS, while maintaining similar performance on tasks without complex source overlap (e.g. ASR, SID). This supports the claim that token mixing improves robustness specifically under complex sound scenes. We agree that further investigation on audio-domain tasks would also be valuable, and we will highlight this as important future work.
>
> ### Q1: Computational cost for MVQ training
> The computational cost of training the MVQ quantiser is relatively small. In SPEAR, we sample 100 hours of data from the pre-training corpus, extract teacher embeddings once, and then train the MVQ quantiser on those embeddings. In our setup, training a 16-codebook MVQ quantiser on 100 hours of data converges after roughly 50k updates, requiring approximately 15 minutes on a single NVIDIA A100 GPU.
>
> ### Q2: Computational cost comparison
> Like WavLM (or HuBERT), SPEAR follows a similar overall pipeline. First, a quantiser is trained on sampled data (k-means for HuBERT/WavLM, MVQ for SPEAR). Second, the teacher model is run over the training set and the quantiser is used to generate discrete training labels, which are stored offline before pre-training begins. Finally, the student model is trained using these pre-computed labels; this is the dominant cost in the overall pipeline.
>
> We agree that SPEAR introduces additional training overhead, since it uses two teachers and therefore requires extra time for label preparation. However, this is a one-time offline cost, and the labels can be reused across future experiments. For our dual-domain SA-97k setup, an estimated time breakdown is below:
> | Method | Quantiser Training Time | Label preparation time (GPU hours) | Pre-training time |
> | --- | --- | --- | --- |
> | WavLM/HuBERT style | ~1h | ~500 h | ~1900 h |
> | SPEAR$_{s+a}$ Large | ~2h | ~580 h | ~2000 h |
>
> As can be seen, SPEAR requires only slightly more time for target preparation. Because SPEAR uses the asymmetrical training strategy, we do not compute audio-teacher labels for speech data, which helps keep the additional label preparation cost modest. The pre-training stage itself is also only slightly longer (approximately 5%) due to the multiple prediction heads for each MVQ codebook.

---

> > ### Author Rebuttal · Reviewer_qmYp · 2026-04-03
> >
> > Thank you for the updates. The authors have clarified most of my concerns, especially w.r.t. computation costs. I encourage the authors to emphasize this more in future versions of the manuscript. Comparison of the versions of SPEAR trained on WavLM vs HuBERT on non-speech audio tasks will be beneficial as well.
> >
> > In general, there are still fundamental limitations with the proposed method, such as the reliance on teacher models and the somewhat expected outcome of the students outperforming if not equaling the teachers. However, I think the empirical results are strong enough that the paper would be of interest to members of the community. I will therefore maintain my score of accept.

---

> > > ### Author Response · Authors · 2026-04-03
> > >
> > > Thank you very much for the thoughtful follow-up and for maintaining your Accept recommendation of score 5. We are very glad to hear that our rebuttal has resolved most of your concerns, and that you believe SPEAR would be of interest to the community.
> > >
> > > We also appreciate your suggestions regarding the manuscript. In the revision, we will make the discussion of computational cost more explicit, and we will also expand the discussion around WavLM-vs-HuBERT teacher choices on non-speech audio tasks, as you suggested. Regarding the remaining limitations, we agree that reliance on teacher models is a trade-off of the framework. Our aim is not to remove teacher dependence, but to show that strong domain-specific speech and audio teachers can be unified effectively within a single encoder. We will make this motivation and limitation clearer in the revised manuscript.
> > >
> > > Thank you again for the encouraging assessment and constructive feedback.

---

### Official Review · Reviewer_8BVK · 2026-03-13

**Soundness:** 3
**Presentation:** 2
**Significance:** 3
**Originality:** 3
**Overall Recommendation:** 4
**Confidence:** 4

**Summary:**

This paper proposes a unified SSL framework that learns speech and audio representations. Generally speaking, speech representations are used to process the spoken content, and audio representations are used to process the acoustic surface of the input audio. This paper attempts to unify the two different types of SSL (or token) to realize a general-purpose speech and audio representation. Technically speaking, multi-codebook vector quantization (MVQ) is applied to speech and audio, which provide a teacher model for speech and another teacher model for audio. Knowledge distillation is applied for both the models to produce the target and unified representation. Evaluations were conducted using ASR, AT, SUPERB, and HEAR. Higher performances were realized compared to some baseline systems.

**Compliance With Llm Reviewing Policy:**

Affirmed.

**Final Justification:**

My first judgment was biased because of my misunderstanding (or authors' inadequate description), but they will clarify that point in the next version of paper. So, I adjusted my evaluation.

**Key Questions For Authors:**

Why "generation" tasks are avoided to test the proposed method?

**Limitations:**

In ICML papers, limitations are generally discussed in a separate section. In this paper, however, a limitation section is not found.

**Strengths And Weaknesses:**

This paper is strong in that higher performances are realized in the various tasks that are tested.
However, the reviewer has to say that this paper is very weak in that almost all the tasks are basically "classification" tasks, and that "generation" tasks are rare. In introduction, the authors seem to pursue "general-purpose" models. In this case, the downstream tasks have to be well-balanced between "classification" tasks and "generation" tasks. In SUPERB, only ST, SE, SS, and VC are "generation" tasks and in HEAR, all the tasks are "classification" tasks. This means that the downstream tasks adopted in this paper are strongly biased and not well prepared to discuss general-purpose representations or models. For example, speech reconstruction, speaker ID conversion, emotion conversion, accent conversion, etc should be used to test the proposed method.

In Table 14, not good performances are obtained in the generation tasks. To realize higher performances in the above generation tasks, speech representations may have to be obtained so that they can capture linguistic, para-linguistic, and extra-linguistic aspect of speech separately. The reviewer is not sure whether the proposed method (Figure 1) can divide input speech or audio stream into multiple aspects. It is well-known that RVQ can separate the input speech stream into a linguistic stream and an extra-linguistic stream to some degree, but it seems that MVQ cannot.

---

> ### Author Rebuttal · Authors · 2026-03-31
>
> We thank the reviewer for acknowledging the strong performance of SPEAR on ASR, AT, SUPERB, and HEAR. We believe, however, that there is a key misunderstanding regarding the term "general-purpose" in our paper.
>
> ### Clarification of "general-purpose"
> In the context of SPEAR, “general-purpose” refers to a unified representation space that generalises well across both speech and general audio understanding tasks, rather than a model that jointly supports both understanding and generation. As stated in the abstract and introduction, the goal of SPEAR is to **bridge the gap between speech understanding and audio understanding, since existing SSL representation models are typically specialised for only one of these domains.**
>
> Accordingly, our evaluation focuses on whether the learned representation generalises across a broad range of speech and audio tasks, which is exactly what ASR, AT, SUPERB, and HEAR are designed to measure. These are standard and widely used benchmarks for evaluating speech and audio representations. Our claim is therefore that SPEAR is a general-purpose representation model for unified speech and audio understanding, not a universal model for all classification, generation, and synthesis tasks. **We will further clarify this in the next version of our paper.**
>
> We would also like to emphasise that **unifying speech and general audio within a single representation space is already an important problem**. Existing SSL models are largely domain-specific, and strong performance does not transfer automatically from speech to audio or vice versa. For example in Table 3, the speech-only model performs well on ASR but poorly on AT, whereas the audio-only model performs much worse on ASR. Because the two domains place different demands on the representation, learning a single encoder working well across both is itself a meaningful advance towards a comprehensive audio perception capability in AI systems.
>
> ### Why generation is not the focus of this paper
> We would also like to emphasise that generation is not the focus of this paper, because SPEAR is an encoder-only SSL framework. It does not have a generative decoder or a codec bottleneck, and is therefore not designed as a generative model. Evaluating tasks such as speech reconstruction or voice conversion would depend heavily on the downstream decoder design and training recipe, and would therefore not isolate the contribution of the learned representation itself. For this reason, we limit the inclusion of generation-related evaluations, in order to maintain a clear and focused assessment of the core contributions of the paper.
>
> ### Clarification on Table 14 and disentanglement
>
> That said, generation-related tasks are not absent from our evaluation. In Table 14, we report results on speech enhancement (SE), speech separation (SS), and voice conversion (VC), which are the generation tasks available in SUPERB. **SPEAR outperforms WavLM, the prior SOTA on SUPERB SE and SS by a clear margin**. Regarding VC, we note in the paper that the results are not directly comparable to earlier SSL models because the SUPERB VC recipe has changed (page 16, last paragraph).
>
> We would also like to clarify that explicit disentanglement of linguistic, paralinguistic, and extra-linguistic factors is not the objective of SPEAR. In SPEAR, MVQ is used to generate fine-grained discrete targets for masked-prediction pre-training, rather than as a factorised generative codec. Specifically, we apply MVQ to the representations of domain-specific teachers and use the resulting tokens to train an encoder-only representation model. This is a different goal from RVQ-based generative approaches and we believe it should be evaluated accordingly.
>
> Finally, although SPEAR is not explicitly designed for factorising speech into multiple aspects (e.g. linguistic and paralinguistic), it still demonstrates very strong performance on paralinguistic tasks such as speaker verification, speaker diarisation, and emotion recognition (Table 4). These tasks require rich speaker and prosodic information, and the fact that SPEAR performs strongly on them indicates that it captures linguistic and paralinguistic cues effectively. We also provide an analysis of the MVQ feature space in Appendix F.4.2, where we show that a single codebook retains substantial speaker information.
>
> In summary, we believe the reviewer’s concern arises from a broader interpretation of "general-purpose" than the one intended in our paper. Our claim is specifically about domain generality across speech and general audio understanding, rather than universality across both understanding and generation. We thank the reviewer for pointing out this potential ambiguity: the final manuscript will clarify that SPEAR is a general-purpose representation model across speech and audio domains, rather than a universal model for both understanding and generation. **We will include a specific limitations section in the final manuscript.**

---

> > ### Author Rebuttal · Reviewer_8BVK · 2026-04-04
> >
> > Thank you for your comments to my reviews. It seems that I misunderstood what "general-purpose" indicates in this paper, but I require the authors to clarify the scope of this paper in the new version.

---

> > > ### Author Response · Authors · 2026-04-05
> > >
> > > Thank you very much for the thoughtful follow-up and for engaging so carefully with our rebuttal. We are glad that our clarification helped resolve the misunderstanding regarding the term "general-purpose" in SPEAR. We really appreciate your decision to **update the overall recommendation to 4 towards acceptance** after reconsidering our paper.
> > >
> > > We sincerely appreciate your constructive engagement throughout the rebuttal process. Your comments have been very helpful in identifying where the scope of the paper could be communicated more clearly, and we will make sure this is clarified in the revised version.

---

### Decision · Program_Chairs · 2026-04-30

**Decision:**

Accept (regular)

**Comment:**

Overview of the paper:

This paper proposes SPEAR, a unified self-supervised learning framework designed to learn joint representations for both speech and general audio. The method distills knowledge from domain-specific teacher models into a single student model by employing multi-codebook vector quantization (MVQ) to generate discrete training targets for a masked token prediction task. To handle inherent domain mismatch and complex acoustic scenes, the framework incorporates an asymmetrical dual-domain training objective alongside a token mixing augmentation mechanism. Empirical evaluations demonstrate consistent performance across various speech and audio understanding benchmarks, including SUPERB and HEAR.

Strengths:

- The framework introduces a logical and effective combination of distillation, multi-codebook vector quantization, and masked language modeling to bridge the gap between speech and non-speech audio representation learning.
- Reviewers highlighted that the proposed asymmetrical dual-domain objective and the stochastic token mixing mechanism are well-motivated and address complex, overlapping sound environments.
- The empirical evaluations are comprehensive, demonstrating solid performance improvements on a wide range of downstream classification and understanding benchmarks.
- The authors provided thorough ablation studies and controlled experiments that adequately justify the framework's architecture, design choices, and hyperparameter selections.

Areas for improvement:

- The initial manuscript lacked clarity regarding the scope of the term "general-purpose," as the framework is heavily biased toward classification and understanding tasks rather than generation tasks. The scope needs to be clearly defined in the final text.
- The conceptual distinctions and advantages over prior distillation-based methods (such as USAD) and earlier applications of MVQ must be more explicitly articulated in the main claims to properly frame the paper's specific contributions.
- The manuscript would benefit from a dedicated limitations section that explicitly discusses the trade-offs of teacher dependence, the multi-source distillation process, and the associated computational costs.